# RECURRENT MODEL-FREE RL IS A STRONG BASELINE FOR MANY POMDPS

## ABSTRACT

Many problems in RL, such as meta RL, robust RL, and generalization in RL, can be cast as POMDPs. In theory, simply augmenting model-free RL with memory, such as recurrent neural networks, provides a general approach to solving all types of POMDPs. However, prior work has found that such *recurrent model-free RL* methods tend to perform worse than more specialized algorithms that are designed for specific types of POMDPs. This paper revisits this claim. We find that a careful architecture and hyperparameter decisions yield a recurrent model-free implementation that performs on par with (and occasionally substantially better than) more sophisticated recent techniques in their respective domains. We also release a simple and efficient implementation of recurrent model-free RL for future work to use as a baseline for POMDPs.[1]

## 1 INTRODUCTION

While reinforcement learning (RL) is often cast as the problem of learning a single fully observable task, also known as MDP, training and testing on that same task, most real-world applications of RL demand some degree of transfer and partial observability. For example, visual navigation (Zhu et al., 2017) requires adaptation to unseen scenes with occlusion in observations, and human-robot collaboration requires that robots infer the intentions of human collaborators. (Chen et al., 2018).

Many subareas in RL study problems that are special cases of POMDPs, and we summarize them in Table 1. For example, meta RL (Duan et al., 2016; Schmidhuber, 1987; Thrun & Pratt, 2012; Wang et al., 2017) is a POMDP where certain aspects of the reward function or (less commonly) dynamics function are unobserved but held constant through one episode. The robust RL problem (Bagnell et al., 2001; Pattanaik et al., 2018; Pinto et al., 2017; Rajeswaran et al., 2017a) assumes that certain aspects of the dynamics or reward function are

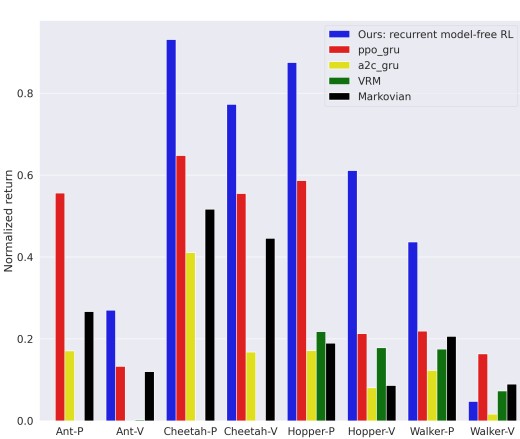

Figure 1: **Implementation Matters for Recurrent Model-Free RL.** This paper identifies critical design decisions for recurrent model-free RL that outperforms not only prior implementations (e.g. PPO-GRU and A2C-GRU from Kostrikov (2018)), but also purpose-designed methods (e.g. VRM from Han et al. (2020)). We also show Markovian policies as lower bounds for reference. The y-axis is **normalized return** given the return of oracle policy (Raffin et al., 2021).

unknown, aiming at finding optimal policies that perform against adversarially-chosen perturbations. Generalization in RL (Cobbe et al., 2019; Packer et al., 2018; Whiteson et al., 2011; Zhang et al., 2018a) focuses on unobserved aspects of the dynamics or reward function that are novel during testing, using an average-case objective instead of a worst-case objective like robust RL. Recent work has proposed efficient and performant algorithms for solving these *specialized* problem settings. However, these algorithms often make assumptions that preclude their application to other classes of POMDPs. For example, methods for robust RL are rarely used for the meta RL setting due to objective mismatch; methods for meta RL are rarely used for general POMDPs due to the stationarity assumption in meta RL.

---

[1]Code: https://drive.google.com/drive/folders/1I5mLlKPf2Gmdpm0nzy9OkR494nCJll1g?usp=sharing

Nonetheless, many prior works have used a simple baseline that is applicable to all POMDPs: model-free RL equipped with a recurrent policy and (sometimes) value function (Duan et al., 2016; Fakoor et al., 2020; Igl et al., 2018; Packer et al., 2018; Rakelly et al., 2019; Wang et al., 2017; Yu et al., 2019). We will refer to this approach as *recurrent model-free RL*. This baseline is simultaneously simple (requiring changing only a few lines of code from a model-free RL algorithm) and general. However, prior work has consistently found that recurrent model-free RL performs poorly across a wide range of problem settings, including meta RL (Rakelly et al., 2019; Zintgraf et al., 2020), general POMDPs (Han et al., 2020; Igl et al., 2018), robust RL (Zhang et al., 2021), and generalization in RL (Packer et al., 2018). One common explanation is that specialized algorithms that are tailored to specific types of POMDPs are very likely to outperform recurrent model-free RL because they (implicitly) encode inductive biases for solving these specific tasks. For example, algorithms for meta RL may leverage the assumption that the underlying dynamics (while unknown) are fixed, and the underlying goals are fixed within one episode (Rakelly et al., 2019; Zintgraf et al., 2020); algorithms for robust RL may assume that the dynamics parameters are known (Rajeswaran et al., 2017a) and dynamics is Lipschitz continuous (Jiang et al., 2021).

This paper challenges this explanation. We argue that, contrary to popular belief, recurrent model-free RL is competitive with recent state-of-the-art algorithms across a range of different POMDP settings. Similar to prior work in Markovian on-policy RL methods (Andrychowicz et al., 2021; Engstrom et al., 2020), our experiments show that implementation in recurrent model-free RL matters. Fig. 1 shows a typical scenario in PyBullet occlusion environments (Coumans & Bai, 2016) to support this argument. Through extensive experiments, we show that the careful design and implementation of recurrent model-free RL is critical to its performance. Design decisions, such as the actor-critic architecture, conditioning on previous actions and/or rewards, the underlying model-free RL algorithms, and context length in RNNs, are especially crucial.

The main contribution of this paper is a performant implementation of recurrent-model free RL. We demonstrate that simple yet important design decisions, such as the underlying RL algorithm and the context length, yield a recurrent model-free RL algorithm that performs on par with prior specialized POMDP algorithms *on the environments those algorithms were designed to solve.* Ablation experiments identify the importance of these design decisions. We also open-sourced our code that is easy to use and memory-efficient.

## 2 BACKGROUND

**MDP.** A Markov decision process (MDP) (Bellman, 1957) is a tuple $(\mathcal{S}, \mathcal{A}, T, T_0, R, H, \gamma)$, where $\mathcal{S}$ is the set of states, $\mathcal{A}$ is the set of actions, $T : \mathcal{S} \times \mathcal{A} \times \mathcal{S} \rightarrow [0, 1]$ is the transition function (dynamics), $T_0 : \mathcal{S} \rightarrow [0, 1]$ is the initial state distribution, $R : \mathcal{S} \times \mathcal{A} \times \mathcal{S} \rightarrow \mathbb{R}$ is the reward function, $H \in \mathbb{N}$ is the time horizon, and $\gamma \in [0, 1)$ is the discount factor. Solving an MDP requires learning a memoryless policy $\pi : \mathcal{S} \times \mathcal{A} \rightarrow [0, 1]$ that maximizes the expected discounted return: $\pi^* = \arg\max_\pi \mathbb{E}_{s_t, a_t, r_t \sim T, \pi} \left[ \sum_{t=0}^{H-1} \gamma^t r_{t+1} \mid s_0 \right]$. For any MDP, there exists an optimal policy that is both memoryless and deterministic (Puterman, 2014). MaxEnt RL algorithms (Ziebart, 2010), such as SAC (Haarnoja et al., 2018a), add an entropy bonus to the RL objective.

**POMDP.** A partially observable Markov decision process (POMDP) (Åström, 1965) is a tuple $(\mathcal{S}, \mathcal{A}, \mathcal{O}, T, T_0, O, O_0, R, H, \gamma)$, where the underlying process is an MDP $(\mathcal{S}, \mathcal{A}, T, T_0, R, H, \gamma)$. Let $\mathcal{O}$ be the set of observations and let $O : \mathcal{S} \times \mathcal{A} \times \mathcal{O} \rightarrow [0, 1]$ be the emission function. Let the observable trajectory up to time-step $t$ be $\tau_{0:t} = (o_0, a_0, o_1, r_1, \ldots, a_{t-1}, o_t, r_t)$, the memory-based policy in *the most general* form is defined as $\pi(a_t \mid \tau_{0:t})$, conditioning on the whole history. At the first time step $t = 0$, an initial state $s_0 \sim T_0(\cdot)$ and initial observation $o_0 \sim O_0(\cdot \mid s_0)$ are sampled. At any time-step $t \in \{0, \ldots, H-1\}$, the policy emits the action $a_t \in \mathcal{A}$ to the system, the system updates the state following the dynamics, $s_{t+1} \sim T(\cdot \mid s_t, a_t)$, then the next observation is sampled $o_{t+1} \sim O(\cdot \mid s_{t+1}, a_t)$ and the reward is computed as $r_{t+1} = R(s_t, a_t, s_{t+1})$.

We refer to the part of the state $s_t$ at current time-step $t$ that can be directly unveiled from *current* observation $o_t$ as the *observable state* $s_t^o$, and the rest part of the state as the *hidden state* $s_t^h$. We call the hidden state $s_t^h$ *stationary* if it does not change within an episode. In this scenario, the policy objective can be rewritten as $\pi^* = \arg\max_\pi \mathbb{E}_{s^h \sim T_0} \left[ \mathbb{E}_{s_t, a_t, r_t \sim T, O, O_0, \pi} \left[ \sum_{t=0}^{H-1} \gamma^t r_{t+1} \mid s^h \right] \right]$ for the average-case POMDP objec-

Table 1: **The summary of selected POMDP subareas.** For each subarea, we list the information of the hidden state $s^h$ including its appearance in dynamics and reward function and its stationarity during one trajectory. We also list the policy input space that are connected with the hidden states, where o, a, r, and d refer to the sequence of observations, actions, rewards, and done signals, respectively. Finally, we list the RL objective in terms of average-case or worst-case, and whether there is a domain shift between training and testing environments. We append the check (✓) or cross mark (✗) with * if it applies to some but not all the work in that subarea. The notation in this table will be covered in Sec. 2.

| Subarea | $s^h$ in dynamics? | $s^h$ in reward? | Is $s^h$ stationary? | Policy input space | RL objective | Domain shift? |
|---|---|---|---|---|---|---|
| "Standard" POMDP | ✓ | ✓ | ✗ | oar | Avg | ✗ |
| Meta RL | ✗* | ✓ | ✓ | oard | Avg | ✗ |
| Robust RL | ✓* | ✗* | ✓* | oa | Worst | ✗ |
| Generalization in RL | ✓* | ✗* | ✓* | oa | Avg | ✓* |

tive, or $\pi^* = \arg\max_\pi \min_{s^h \in \text{supp}(T_0)} \mathbb{E}_{s_t, a_t, r_t \sim T, O, O_0, \pi} \left[ \sum_{t=0}^{H-1} \gamma^t r_{t+1} \mid s^h \right]$ for the worst-case POMDP objective.

## 3 RELATED WORK

In this section, we discuss several subareas of RL that both explicitly and implicitly solve POMDPs, as well as algorithms proposed for these specialized settings. Table 1 summarizes these subareas.

**RL for "Standard" POMDPs.** We use the term "standard" to refer to prior work that explicitly labels the problems studied as POMDPs. Common tasks include scenarios where the states are partially occluded (Heess et al., 2015), different states correspond to the same observation (perceptual aliasing (Whitehead & Ballard, 1990)), random frames are dropped (Hausknecht & Stone, 2015), observations use egocentric images (Zhu et al., 2017), or the observations are perturbed with random noise (Meng et al., 2021). These POMDPs often have hidden states that are non-stationary and affect both the rewards and the dynamics. POMDPs are hard to solve (Littman, 1996; Papadimitriou & Tsitsiklis, 1987) because of the curse of dimensionality: the size of the history grows linearly with the horizon length. Many prior POMDP algorithms (Cassandra et al., 1994; Kaelbling et al., 1998) attempt to infer the state from the past sequence of observations, and then apply standard RL techniques to that inferred state. However, the exact inference requires the knowledge of the dynamics, emission, and reward functions, and is intractable in all except the most simple settings. A common strategy for solving these general POMDPs is to use recurrent policies, which take the entire history of past observations as inputs (Bakker, 2001; Schmidhuber, 1991; Wierstra et al., 2007). This strategy is very simple and general, and can be applied to arbitrary tasks without knowledge of the task structure (e.g., whether the hidden states change within an episode) across long time horizons (Duan et al., 2016). These recurrent strategies can be further subdivided into model-free methods (Hausknecht & Stone, 2015; Heess et al., 2015; Meng et al., 2021) , where the single objective is to maximize the return, and model-based methods (Freeman et al., 2019; Han et al., 2020; Igl et al., 2018; Watter et al., 2015) that have explicit objectives on modeling the belief states and use them as the inputs of memoryless policies. The recurrent model-free RL that we focus on belongs to the class of model-free off-policy memory-based algorithms.

**Meta RL.** Meta RL, also called "learning to learn" (Schmidhuber, 1987; Thrun & Pratt, 2012), focuses on POMDPs where some parameters in the rewards or (less commonly) dynamics are varied from episode to episode, but remain fixed within a single episode, which represent different *tasks* with different values (Humplik et al., 2019). The meta RL setting is almost the same as multi-task RL (Wilson et al., 2007; Yu et al., 2019), but differs in that multi-task RL can observe the task parameters, making it an MDP instead of a POMDP. Algorithms for meta RL can be roughly categorized based on how the adaptation step is performed. Gradient-based algorithms (Fakoor et al., 2020; Finn et al., 2017; Hochreiter et al., 2001) run a few gradient steps on the pre-trained models to adapt. Memory or context-based algorithms use RNNs to implicitly adapt, which can be further subdivided into implicit and explicit task inference methods. Implicit task inference methods (Duan et al., 2016; Wang et al., 2017) use RL objective only to learn recurrent policies. Explicit task inference methods (Rakelly et al., 2019; Zintgraf et al., 2020) train an extra inference model to explicitly estimate task embeddings (i.e., a representation of the unobserved parameters) by variational inference. Task embeddings are then used as additional inputs to memoryless policies.

**Robust RL.**   The goal of robust RL is to find a policy that maximizes returns in the worst-case environments. Early work in the control and operations research community (Khalil et al., 1996; Nilim & Ghaoui, 2005) and RL community (Bagnell et al., 2001; Morimoto & Doya, 2005) focused on linear or finite systems. Prior work designs deep RL algorithms that are robust against a variety of adversarial attacks, including attacks on the dynamics (Jiang et al., 2021; Rajeswaran et al., 2017a), observations (Huang et al., 2017; Pattanaik et al., 2018; Zhang et al., 2021), and actions (Gleave et al., 2020; Pinto et al., 2017; Tessler et al., 2019). Treating the robust RL problem as a POMDP, rather than an MDP (as done in most prior work), unlocks a key capability for RL agents, because agents can use their memory to identify the hidden states of the current adversarial environment, although previous work (Jiang et al., 2021; Rajeswaran et al., 2017a) only train Markovian policies on POMDPs. While some work find memory-based policies are more robust to the adversarial attacks than Markovian policies (Russo & Proutière, 2021; Zhang et al., 2021), they train these baselines in a single MDP without adversaries, which differs from our training setting where the recurrent model-free RL can have access to a set of MDPs.

**Generalization in RL.**   The goal of generalization in RL is to make RL algorithms perform well in test domains that are unseen during training, which emphasizes the average case on the novel test domains instead of the worse case in the possibly seen test domains as in robust RL. Prior work have studied generalization to initial states in the same MDP (Rajeswaran et al., 2017b; Whiteson et al., 2011; Zhang et al., 2018b), random disturbance in dynamics (Rajeswaran et al., 2017b), states (Stulp et al., 2011), observations (Song et al., 2020; Zhang et al., 2018a), and actions (Srouji et al., 2018), and different modes in procedurally generated games (Cobbe et al., 2019; Farebrother et al., 2018; Justesen et al., 2018). Among them, Packer et al. (2018) provides a benchmark on both in-distribution (ID) and out-of-distribution (OOD) generalization to different dynamics parameters, and Zhao et al. (2019) extends the benchmark by introducing random noise in states, observations, and actions. Algorithms for improving generalization in RL can be roughly divided into classic regularization methods such as weight decay, dropout, batch normalization, and entropy regularization (Cobbe et al., 2020; Farebrother et al., 2018; Igl et al., 2019), model architectures (Raileanu & Fergus, 2021; Srouji et al., 2018), data augmentation through randomization (Lee et al., 2020; Tobin et al., 2017), Although introducing observational noise and the change in dynamics parameters will transform MDPs to POMDPs, few work study memory-based policies such as model-free recurrent RL with mixed results. Same algorithm RL2 (Duan et al., 2016) was found to perform badly in Packer et al. (2018) but relatively well in Yu et al. (2019).

## 4   DESIGN CONSIDERATIONS FOR RECURRENT MODEL-FREE RL

Implementing a recurrent model-free RL algorithm requires making a number of design decisions. This section describes the decisions that we found most important to make recurrent model-free RL competitive with more complex, recent algorithms. We will focus on continuous control problems with state-based inputs (i.e., not image-based inputs). Importantly, we assume that the policy can observe the *reward* and *done* signals (the end of one episode during one trial (Duan et al., 2016)) from the environment during *evaluation*. This assumption is common in prior work (Han et al., 2020; Zintgraf et al., 2020), but many recurrent model-free implementations do not provide the agent with information. In the following paragraphs, we will describe the important decision factors in recurrent model-free RL. Table 2 summarizes how prior work and our method makes these design decisions when implementing recurrent model-free RL.

**Recurrent Off-Policy Actor-Critic Architecture.**   The first important design decision is whether the recurrent policy (actor) and the recurrent Q-value function (critic) use *shared* RNN encoder (and embedders) or use *separate* ones. In the experiment section (Sec. 5.2) we will show that a shared encoder would cause large gradient norm in the (off-policy) recurrent actor-critic and thus hinder learning, while separate encoders can greatly mitigate this issue and learn efficiently. This echoes prior work (Fakoor et al., 2020; Meng et al., 2021; Sun et al., 2021; Wang et al., 2020) that also use separate encoders in their (off-policy) recurrent actor-critic. To avoid running an inordinate number of experiments, we will use the *separate* architecture in the rest of the paper.

**Policy Input Space.**   The next consideration is the input space of the model-free policy. The maximal input space of policy to emit an action $a_t$ at time $t$, should be the history of all quantities that the policy has observed, namely the past observations $o_{0:t}$, the past actions $a_{0:t-1}$, the past

Table 2: **How the prior work and our method implement the recurrent model-free RL as their own method or baseline.** We can see that none of the prior work share the same set of decision variables, some of which have bad choices that may lead to the poor performance reported in the prior work. Our method covers a range of choices in these decision factors and finds the combinations in the last rows that lead to **the best performance in terms of the average performance** across the experimented environments in each subarea.

| Algorithm | Domain | Arch | Encoder | Inputs | Len | RL |
|---|---|---|---|---|---|---|
| Duan et al. (2016) | Meta RL | separate | GRU | `oard` | 1000 | TRPO, PPO |
| Wang et al. (2017) | Meta RL | shared | LSTM | `oart` | 5-150 | A2C |
| Baseline from Rakelly et al. (2019) | Meta RL | separate | GRU | `oard` | 100 | PPO |
| Baseline from Zintgraf et al. (2020) | Meta RL | separate | GRU | `oard` | Max | A2C, PPO |
| Baseline from Fakoor et al. (2020) | Meta RL | separate | GRU | `oar` | 10-25 | TD3 |
| Baseline from Yu et al. (2019) | Meta RL | separate | GRU | `oard` | 500 | PPO |
| Kostrikov (2018) | POMDP | shared | GRU | `o` | 5-2048 | PPO, A2C |
| Wang et al. (2020) | POMDP | separate | LSTM | `oa` | 150 | TD3, SAC |
| Meng et al. (2021) | POMDP | separate | LSTM | `oa` | 1-5 | TD3 |
| Yang & Nguyen (2021) | POMDP | separate | both | `oa` | Max | TD3, SAC |
| Baseline from Igl et al. (2018) | POMDP | shared | GRU | `oa` | 25 | A2C |
| Baseline from Han et al. (2020) | POMDP | shared | LSTM | `o` | 64 | SAC |
| Baseline from Zhang et al. (2021) | Robust RL | separate | LSTM | `o` | 100 | PPO |
| Baseline1 from Packer et al. (2018) | Generalization | shared | LSTM | `o` | 128-512 | PPO, A2C |
| Baseline2 from Packer et al. (2018) | Generalization | separate | LSTM | `oard` | 128-512 | PPO, A2C |
| Our method | Meta RL | separate | LSTM | `oard` | 64 | TD3 |
| Our method | POMDP | separate | GRU | `oa` | 64 | TD3 |
| Our method | Robust RL | separate | LSTM | `o` | 64 | TD3 |
| Our method | Generalization | separate | LSTM | `o` | 64 | TD3 |

rewards $r_{0:t}$, and the past done signals $d_{0:t}$, which was already employed in the early work (Duan et al., 2016). Generally, the input space of optimal policy should only depend on the quantities that have connections with *hidden states* (defined in Sec. 2) (Izadi & Precup, 2005; Poupart & Boutilier, 2002). We show the policy input spaces that are connected with the hidden states for the discussed subareas in the "Inputs" column of Table 1. While prior work often only conditions the recurrent RL baseline on previous observations (and actions) (Han et al., 2020; Igl et al., 2018; Kostrikov, 2018; Meng et al., 2021; Wang et al., 2020; Yang & Nguyen, 2021), our experiments in Sec. 5.2 find that additionally conditioning on other previous information, such as previous rewards, can increase reward by up to 30%.

**Model-free RL Algorithms.** Recurrent model-free RL can be understood as applying an off-the-shelf model-free RL algorithm with an actor and a Q function parametrized to take sequences of inputs. As such, the choice of the underlying model-free RL algorithm is paramount. Most prior work on continuous control POMDP problems used *on-policy* algorithms, such as A2C (Mnih et al., 2016), TRPO (Schulman et al., 2015) or PPO (Schulman et al., 2017). While *off-policy* algorithms such as TD3 (Fujimoto et al., 2018) and SAC (Haarnoja et al., 2018a;b) greatly improve the performance in continuous control MDP problems in terms of sample efficiency and asymptotic performance, these methods are rarely used in recurrent model-free RL baselines (Rakelly et al., 2019; Zhang et al., 2020; Zintgraf et al., 2020). In the experiment section (Sec. 5.1), we will show that using these off-policy algorithms for recurrent model-free RL provides results that are better than using on-policy algorithms and are comparable to their specialized methods in POMDP. This echoes the finding that model-free off-policy TD3-Context (Fakoor et al., 2020) can be better than the specialized method PEARL (Rakelly et al., 2019) in meta RL.

**RNN Variants and Context Length.** RNN training is known to be unstable, especially with long sequences input (Bengio et al., 1994). The RNN variants like LSTM (Hochreiter & Schmidhuber, 1997) and GRU (Chung et al., 2014) mitigate the training issues, but still may fail to learn long-term dependencies (Trinh et al., 2018). In POMDP problems, these dependencies reflect the memory that an agent must have to solve a task. For example, a POMDP that hides velocities from observations theoretically requires a short memory length to infer velocities through consecutive positions (Meng et al., 2021). Prior work in POMDPs choose a variety set of context lengths for RNNs from 1 to 2048 (see the "Len" column of Table 2), and we select three representatives of short (5), medium (64), and long length (larger than 100) in the experiments (Sec. 5) for comparison. We also try both LSTM and GRU as RNN variants to compare their performance. We find that the optimal context length and RNN variant are task-specific (see Sec. 5.2).

## 5 EXPERIMENTS

Our experiments aim to answer two questions. First, how does a *well-tuned* implementation of recurrent model-free RL compare to specialized POMDP methods, such as purpose-designed meta RL and robust RL algorithms? To give these prior methods the strongest possible footing, we will compare prior methods on the specific problem types for which they were developed (i.e., meta RL algorithms were tested on meta RL tasks). Our second question studies which design decisions are essential for recurrent model-free RL. We put the environment details in Appendix D.

**Code Implementation.** We release a modular and highly-configurable implementation of recurrent (off-policy) model-free RL: url. Our implementation is efficient in terms of computer memory compared to previous off-policy RL methods for POMDPs (200x less RAM than Han et al. (2020) and 9x less GPU memory than Dorfman et al. (2020)). Please see the appendix A for details, including an explanation of why our implementation is more memory-efficient than prior work.

### 5.1 RECURRENT MODEL-FREE RL IS COMPARABLE WITH PRIOR SPECIALIZED METHODS

While prior work has studied a range of different POMDP settings (e.g., meta RL, occluded observations), recurrent model-free RL is a ubiquitous baseline (Han et al., 2020; Humplik et al., 2019; Igl et al., 2018; Rakelly et al., 2019; Zintgraf et al., 2020). However, prior work consistently report that this baseline is reported to be unperformed to more specialized methods. This section casts doubt on that claim, showing that a well-tuned implementation of recurrent model-free RL can perform *at least* as well as more specialized methods.

We study four subareas of POMDPs: the "standard" POMDP, meta RL, robust RL, and generalization in RL. We tune a wide range of decision factors shown in Sec. 4 in our implemented recurrent model-free RL. Appendix A.3 shows the details of the tuning options. For each subarea, we show the performance of **a single variant** that works best across the environments in that subarea, compared with the prior specialized methods in this subsection. In other words, the following plots of each subarea report **the same model-free recurrent RL algorithm** with the same hyperparameters. The exact configurations of each subarea can be found in the last four rows of Table 2. Under this restricted setting, we find that our implementation can actually outperform prior (specialized) methods by a wide margin across the four subareas.

For each plot of learning curves, we show three approaches as reference. First, an **Oracle** policy has access to the POMDP hidden states, turning the POMDP into an MDP; this policy should therefore be treated as an upper bound on the performance that any POMDP method should receive. Second, as a lower bound, we use a **Markovian** policy to solve the POMDP. Both Oracle policy and Markovian policy are trained with the same hyperparameters as our recurrent model-free RL implementation. Third, we add a **Random** policy, which represents a trivial lower bound. We show the full learning curves in Appendix E.1 due to the space limit. See Appendix B for details about the implementation of these comparisons.

**"Standard" POMDP.** Our first experiments look at the "standard" POMDPs that typically occlude some part of states in the environment. We will compare against **VRM** (Han et al., 2020), a recent state-of-the-art model-based POMDP algorithm. We directly apply the environment design of VRM paper that occludes either positions&angles or velocities of the simulated robot in PyBullet (Coumans & Bai, 2016). There are 8 environments {Hopper, Ant, Walker, Cheetah}-{P,V}, where "-P" stands for observing positions&angles only, and "-V" stands for observing velocities only. Fig. 1 and Fig. 18 in appendix show that the best single variant of our model-free recurrent RL implementation outperform VRM in 6 out of 8 environments, especially in {Cheetah,Hopper}-{P} (over $80\%$ of the Oracles). Our results suggest that, while the variational dynamics model used by VRM may be useful for some tasks, a simple recurrent model-free RL baseline can outperform VRM if properly tuned. While we are primarily interested in sample complexity, but not compute, it is worth noting that our recurrent model-free RL implementation is substantially more efficient than the open-source VRM implementation, training $5\times$ faster and requiring at most $200\times$ less RAM usage (see Appendix A).

**Meta RL.** We next compare the recurrent model-free RL to the meta RL setting, where some indicator of the task is unobserved. We compare our implementation of recurrent model-free RL to a specialized state-of-the-art method, **VariBAD** (Zintgraf et al., 2020) that explicitly learns the

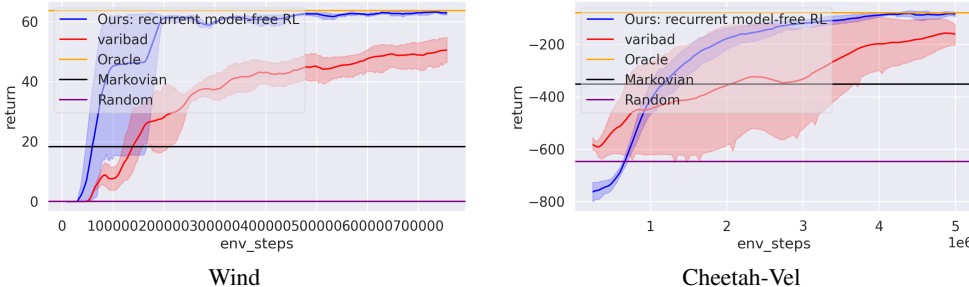

Wind                                Cheetah-Vel

Figure 2: **Learning curves on two meta RL environments.** The **single best variant** of our implementation on recurrent model-free RL can surpass the specialized meta RL method off-policy VariBAD (Dorfman et al., 2020), and match the performance of an "Oracle" policy that gets to observe the hidden state.

task embeddings by variational model-based objectives. As suggested by Dorfman et al. (2020), we modified VariBAD to use SAC instead of PPO. This change also allows us for fair comparison with our implementation of recurrent model-free RL, which uses SAC and TD3. We adopt the three environments used in Dorfman et al. (2020) for experiments, including Semi-Circle and Cheetah-Vel, and we also adapt Wind to make it harder to solve. Figure 2 shows that our best single variant outperforms VariBAD and even reaches Oracles in the two meta RL environments, Cheetah-Vel and Wind, leaving Semi-Circle in the appendix. Prior work (Rakelly et al., 2019; Zintgraf et al., 2020) show that disentangling task inference and control can stabilize training. However, our experiments suggest that joint training of task inference and control could also have comparable performance if well implemented. Additionally, because recurrent model-free RL is trained end-to-end, without using pre-trained task representations saved in the replay buffer like the off-policy VariBAD (Dorfman et al., 2020), our implementation does not have non-stationarity issue in task representations.

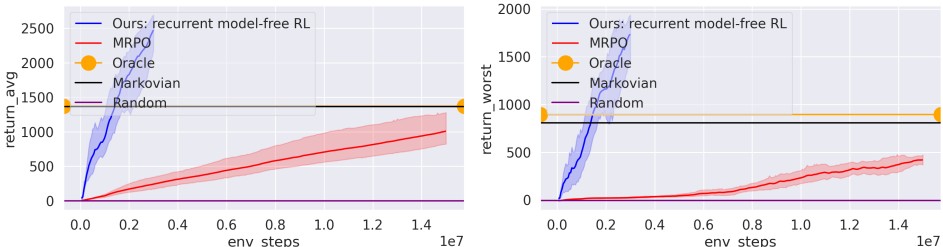

Figure 3: **Learning curves on one robust RL environment,** Cheetah-Robust. We show the average returns (left figure) and worst returns (right figure) of each method. The **single best variant** of our implementation on recurrent model-free RL can greatly outperform the specialized robust RL method MRPO (Jiang et al., 2021), and surpass the Oracle that are trained with same simulation and gradient steps.

**Robust RL.** Thirdly, we focus on the robust RL that aims to maximize the worst returns over the tasks. We choose the recent specialized algorithm **MRPO** (Jiang et al., 2021) as the compared method, and adopt their used environments based on SunBlaze benchmark (Packer et al., 2018). These environments have hidden states that are fixed during one episode, including the density and the friction coefficients of the simulated robots, namely {Cheetah, Hopper, Walker}-Robust. Fig. 3 shows both the average return and worst return of our single best variant and MRPO on the three environments, where the worst return is measured by the average return in the worst $10\%$ testing tasks following the practice in Jiang et al. (2021). The results are quite surprising: although our implementation, using average-case RL objective and *without* access to the hidden states, is not expected to surpass MRPO and Oracle *with* access to hidden states in worst return, we found that our best variant vastly outperforms the specialized MRPO and Oracle in both average return and worst return, with over $80\%$ fewer simulation steps. Our implementation benefits from its memory and off-policy algorithms, while MRPO might suffer from its Markovian on-policy algorithm and a bit ideal Lipschitz assumption in dynamics. Nevertheless, our implementation is around 17.5x slower than MRPO given the same simulation steps (see Appendix A), so we only run it with 3M steps with a limited time budget.

**Generalization in RL.** Finally, we focus on the SunBlaze benchmark from Packer et al. (2018) for investigating generalization in RL, including {Hopper, Cheetah}-Generalize. We pick the best specialized method in the tables of final performance in Packer et al. (2018), Markovian on-policy robust RL method **EPOpt-PPO-FF** (Rajeswaran et al., 2017a). Fig. 4 show the interpolation and

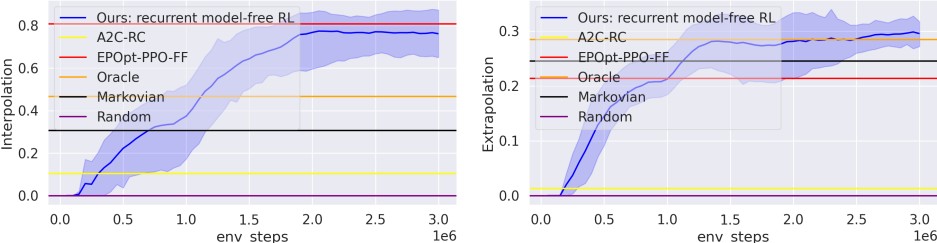

Figure 4: **Learning curves on generalization in one RL environment,** Hopper-Generalize. We show the interpolation success rates (left figures) and extrapolation success rates (right figures) of each method. The **single best variant** of our implementation on recurrent model-free RL can be par with the specialized method EPOpt-PPO-FF (Rajeswaran et al., 2017a) in interpolation and outperform it in extrapolation. The data of EPOpt-PPO-FF and A2C-RC (a recurrent model-free on-policy RL method) are copied from the Table 7 & 8 in Packer et al. (2018).

extrapolation success rates in one environment, where in the interpolation the testing tasks have same distribution of hidden states as that of the training tasks, while in the extrapolation the testing distribution is disjoint from that of training. We can see that our model-free method is on par with the EPOpt-PPO-FF in the interpolation benchmark, while EPOpt-PPO-FF requires access to the dynamics parameters but ours does not. In the extrapolation benchmark, our method greatly outperforms the previous method, although our objective does not consider extrapolation.

Overall, we can see that with careful tuning on recurrent model-free RL, it can at least perform as well as the specialized or more complicated methods, in various kinds of POMDPs.

## 5.2 WHAT MATTERS IN RECURRENT MODEL-FREE RL ALGORITHMS?

In the previous subsection, we showed that recurrent model-free RL can perform on par with the specialized (state-of-the-art) methods, then a natural question comes: Why our implementation of recurrent model-free RL outperforms the implementation used in prior work?

Our analysis will focus on ablating the five important design decisions introduced in Sec. 4: the actor-critic architecture (`Arch`), the policy input space (`Inputs`), the underlying model-free RL algorithm (`RL`), the RNN encoder (`Encoder`), and the RNN context length (`Len`). See Table. 2 for a summary of how prior work made these design decisions. Due to the space limit, we show the ablation results in some but not all the environments to compare the performance between the best single variant and the other variant that only differs in one decision factor. We also provide "single factor analysis" plots for each decision factor by averaging the performance over the other factors in Appendix E.2.

**Recurrent Off-Policy Actor-Critic Architecture.** First, we ran some experiments with both shared and separate architectures on two toy POMDP environments. Fig. 5 show the results in one of them (see Appendix E.3 for the other). We can see that the shared architecture failed to learn, compared to the separate architecture. The large RNN gradient norm in the shared architecture suggests that the actor and critic losses may cause conflicts in gradient update. Our results echo prior work (Fakoor et al., 2020; Meng et al.,

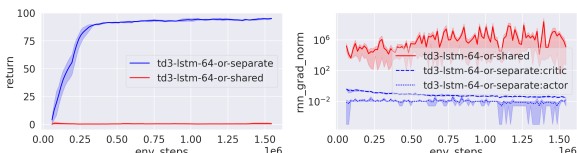

Figure 5: **Comparison between *shared* and *separate* recurrent actor-critic architecture** with all the other hyperparameters same, on Semi-Circle, a toy meta RL environment. We show the performance metric (left) and also gradient norm of the RNN encoder(s) (right, in **log-scale**). For the separate architecture, `:critic` and `:actor` refer to the separate RNN in critic and actor networks, respectively.

2021; Sun et al., 2021) that only consider *separate* RNN encoders that can achieve high asymptotic rewards, and also echo that (Han et al., 2020) shows poor results in the *shared* architecture of SAC-LSTM.

**Policy Input Space.** The 1st row of Table 3 shows the effect of policy input space in a POMDP environment Walker-P. The reward signals could help reveal the missing information of the velocity of the robot base, which is occluded in Walker-P. Similarly, the single factor analysis on policy input in Fig. 12 shows that `oar` is among the best in "-P" environments. Therefore, it is reasonable that adding previous rewards into policy inputs can increase the performance.

Table 3: **Ablation results in our implementation of recurrent model-free RL.** In this table, we show how a single change in one decision factor from the variant that is best on average in that subarea, could significantly increase the performance. The first column shows how we change the single decision factor, and the last column shows the performance comparison between **the best variant in that subarea** (left) and **the ablated one** (right). For robust RL and generalization in RL, we show the performance metric in worst returns and extrapolation success rates, respectively.

| Change in one decision factor | Subarea | Env | Performance comparison |
|---|---|---|---|
| Inputs: oa → oar | "Standard" POMDP | Walker-P | 981.6 → 1345.0 (1.3×) |
| RL: TD3 → SAC | "Standard" POMDP | Ant-P | 310.7 → 2123.5 (6.8×) |
| Encoder: LSTM → GRU | Robust RL | Walker-Robust | 765.9 → 931.3 (1.2×) |
| Len: 64 → 400 | Meta RL | Cheetah-Vel | -85.2 → -74.6 (+14%) |
| Len: 64 → 5 | Generalization | Hopper-Generalize | 0.292 → 0.415 (1.4×) |
| Len: 64 → 5 | "Standard" POMDP | Walker-V | 121.4 → 264.3 (2.2×) |

**Model-free RL Algorithms.** Table 2 shows that TD3 dominates all the best variants in each subarea, which may be due to dense reward setting in most environments. However, the 2nd row of Table 3 shows the effect of RL algorithm in a POMDP environment Ant-P. SAC is significantly better than TD3 (increase by 6.8×, surpassing the PPO-GRU (Kostrikov, 2018) in Fig. 1), possibly due to strategic exploration where the action noise conditions on the history instead of being independent. This is prominent mainly in Ant-P as it might be much harder than the others.

**RNN Variants and Context Length.** Generally, there is no significant difference between LSTM and GRU (see the single factor analysis in Appendix E.2), which is understandable as both are designed for general purpose. Howover, the 3rd row of Table 3 shows the effect of RNN encoder in a robust RL environment. We can see replacing LSTM with GRU can increase the worst-case metric in Walker-Robust. For the context length in RNNs, a medium length (64) dominates in all the best variants in each subarea (see Table 2), which could be viewed as a trade-off between memory capacity and computation costs. However, the remaining rows of Table 3 show the mixed effects of context length in RNNs. Both increasing and decreasing the context length can boost the performance in different environments. Specifically, decreasing the length from 64 to 5 makes our method surpass VRM in Walker-V (increase by 2.2×). This might explain why the prior methods adopt a wide range of context lengths from 1 to 2048 (see Table 2). Therefore, the choice of context length is problem-specific and can require tuning.

**Summary.** We now summarize the main findings of our experiments:

1. Using separate weights for the recurrent actor and recurrent critic boosts performance, likely because it avoids gradient explosion (Fig. 5 and Fig. 17 in Appendix).
2. Using state-of-the-art off-policy RL algorithms as the backbone in recurrent model-free RL can improve asymptotic performance (Fig. 1 and Figures in Appendix E.1).
3. The context length for the recurrent actor and critic has a large influence on task performance, but the optimal length is task-specific. Reasonable values are 5 to 500, and 64 is a good start (Rows 4–6 in Table 3 and Figures in Appendix E.2).
4. It is important that the inputs to the recurrent actor and critic, such as past observations and past returns, contain enough information to infer the POMDP hidden states (Row 1 in Table 3 and Figures in Appendix E.2).

These findings may provide a useful initialization for researchers studying recurrent model-free RL.

## 6 CONCLUSION

In this paper, we show that a carefully-designed implementation of recurrent model-free RL can perform well across a range of POMDP domains, often on par with (if not significantly better than) prior methods that are specifically designed for specific types of POMDPs. Our ablation experiments demonstrate the importance of key design decisions, such as the underlying RL algorithm and RNN context length. While the best choices for some decisions (such as using separate RNNs for the actor and the critic) are consistent across domains, the best choices for other decisions (such as RNN context length) are problem-dependent.

## REPRODUCIBILITY STATEMENT

We release our code at the url for reproducibility. In this code repository, we provide instructions on how to run our method and the compared methods in this paper on all the environments involved in the experiment section. We provide the default configuration files for training and evaluating these algorithms that are adopted in our experiments. We also attach the numeric results of all the bar charts shown in the experiment section.

## ETHICS STATEMENT

We do not believe that our work has direct ethical or societal implications.

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

# A    CODE-LEVEL DETAILS

In this section, we first introduce the outline of code design, especially the replay buffer for sequences, and then compare the system usage, including computing speed, RAM, and GPU memory with previous POMDP methods.

## A.1    CODE DESIGN

**Easy to use.**    Our code can be either used as an API to call the recurrent model-free RL class or a framework to tune the details in the class. The recurrent model-free RL class takes the hyperparameters of RNN encoder type, shared or separate actor-critic architecture, and whether include previous observations, and/or actions, and/or rewards into the inputs, to generate different instances. The details of the hyperparameter tuning set are shown in Sec. A.3.

**Memory-efficient replay buffer for sequences.**    Moreover, we design an efficient replay buffer for off-policy RL methods to cope with sequential inputs. Previous methods (Han et al., 2020; Yang & Nguyen, 2021) mainly use *three-dimensional* replay buffer to store sequential inputs, with the dimensions of (num episodes, max episode length, observation dimension), taking observation storage as example. This kind of implementation becomes memory-inefficient if the actual episode length is far smaller than the max episode length (then there will be many padded zeros (Dorfman et al., 2020)). Instead, we manage to implement a two-dimensional replay buffer of shape (num transitions, observation dimension) for observation storage, which also records the locations where each stored episode ends. In case of actual episodes that are shorter than provided sampled sequence length, the buffer also generates *on-the-fly masks* to indicate if the corresponding transitions are valid, so that we do not need to save zero-padded observations in the buffer. This enables the policy to receive a batch of previous experiences in a tensor-like data structure when sampling from the replay buffer. To sum up, our replay buffer can support varying-length sequence inputs and subsequence sampling without zero padding.

**Flexible training speed.**    Finally, our code supports flexible training speed by controlling the ratio of the numbers of gradient updates in RL w.r.t. the environment rollout steps. The training speed is approximately proportional to the ratio if the simulator speed is much faster than the policy gradient update. Typically, the ratio is less than or equal to 1.0 to enjoy higher training speed.

## A.2    SYSTEM USAGE

Table 4 shows the typical system usage of our method and the compared specialized methods on different environments. The time cost for our method and VariBAD depends on how many processes in parallel are run on a single GPU – our method is run with 8 processes on a single GPU while VariBAD is run with one process due to large GPU memory usage. From the results we can see that our method is memory-efficient in both RAM and GPU, and has an acceptable training speed with default hyperparameters. The computer system we used during the experiments includes a GeForce RTX 2080 Ti Graphic Card (with 11GB memory) and Intel(R) Xeon(R) Gold 6148 CPU @ 2.40GHz (with 250GB RAM and 80 cores).

Table 4: **Comparison between our method and specialized methods in system usage.** The time costs are evaluated within 1M environment steps. Both VRM and MRPO are run on CPUs and MRPO does not have a replay buffer (shown in N/A). VariBAD requires the assumption of fixed episode length for the RAM cost.

| Method | Environment | Time cost | RAM | GPU memory |
|---|---|---|---|---|
| **Ours** (GRU) | Hopper-V | 22.5 h | O(1) | 1.2 GB |
| VRM (Han et al., 2020) | Hopper-V | 102 h | O(200) | N/A |
| **Ours** | Semi-Circle | 12 h | O(1) | 1 GB |
| VariBAD (Dorfman et al., 2020) | Semi-Circle | 2.3 h | O(1)* | 9.5 GB |
| **Ours** | Cheetah-Robust | 7 h | O(1) | 1.1 GB |
| MRPO (Jiang et al., 2021) | Cheetah-Robust | 0.4 h | N/A | N/A |

### A.3 OUR HYPERPARAMETER TUNING SET

Our proposed method has the following decision factors (introduced in Sec. 4) to tune in the experiments with the following options (the names in brackets are abbreviated ones):

- Actor-Critic architecture (**Arch**): share the encoder weights between the recurrent actor and recurrent critic or not, namely `shared` and `separate`.
- Model-free RL algorithms (**RL**): `td3` (Fujimoto et al., 2018) or `sac` (Haarnoja et al., 2018a)
- Encoder architecture (**Encoder**): `lstm` (Hochreiter & Schmidhuber, 1997) or `gru` (Cho et al., 2014)
- Policy input space (**Inputs**): `o`, `oa`, `or`, `oar`, `oard` (the notation is introduced in Sec. 4; depending on the POMDPs, see "Policy input" row in Table 5)
- Context length (**Len**): short (5), medium (64), long (larger than 100, depending on the POMDPs).

For each instance, we label it with the names of all the hyperparameters it used in lowercase as notation. For example, `td3-lstm-64-or-separate` in Fig. 5 refers to the instance that uses the separate actor-critic architecture, TD3 RL algorithm, LSTM encoder, the policy input space of previous observations and reward sequences, and RNN context length of 64.

## B TRAINING DETAILS

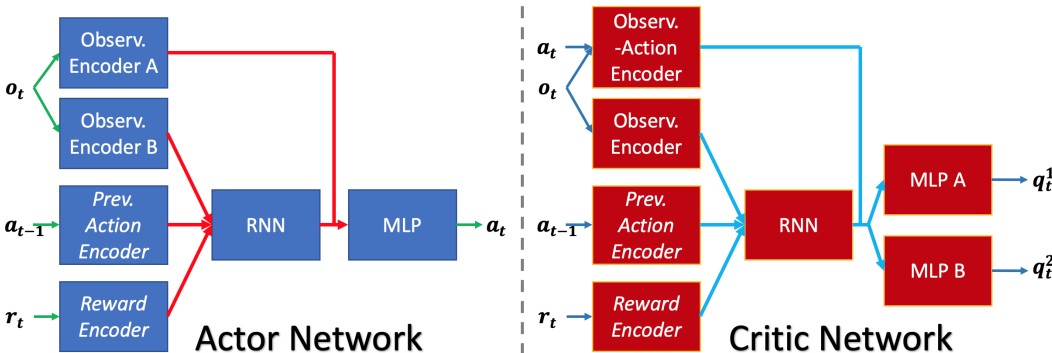

Figure 6: **The network architecture of our implementation on recurrent model-free RL.** The left part shows the actor network, and the right shows the critic network. Each block shows a trainable module, with independent weights. We italicize the previous action and reward encoders as they are optional. By default, each encoder has one hidden layer, each RNN is one-layer LSTM or GRU, each MLP has two hidden layers.

Fig. 6 shows our (separate) recurrent actor-critic architecture. Table 5 shows the main hyperparameters we adopt for each subarea. We did not tune these hyperparameters, except that we adjusted the number of gradient steps so that all the experiments could be completed in 36 hours.

For **Markovian** policies (SAC and TD3), we remove the encoders and RNNs from the actor-critic architecture, and train them with same hyperparameters as those of recurrent policies. For each task, we report the results of *either* SAC or TD3, whichever achieves higher returns. For **Oracle** policies, we use the well-tuned results from Table 1 ("SAC w/ unstructured row") in Raffin et al. (2021), which is based on Stable Baseline3 (Raffin, 2020), for "standard" POMDPs. For the other subareas, we have to run the Markovian policies (SAC and TD3) with access to the hidden states, using the same training hyperparameters as those of recurrent policies. But these Oracle policies might be not well-tuned given same environment and gradient steps, especially in robust RL and generalization in RL.

We also show the settings of the specialized methods we compared in the main paper in Table 6. Note that our recurrent model-free RL share the exactly same settings as (off-policy) VariBAD (Dorfman

Table 5: **Hyperparameter summary in our implementation of model-free recurrent RL.** For each subarea, we report the hidden layer size of each module, RL and training hyperparameters. For Meta-RL, we take the model on Cheetah-Vel as example, which follows the architecture design of off-policy VariBAD (Dorfman et al., 2020). The hidden size of observation-action encoder is the sum of that of observation encoder, previous action encoder (if exists), and reward encoder (if exists).

| | | Meta-RL* | POMDP | Robust RL | Gen. in RL |
|---|---|---|---|---|---|
| Hidden layer size | Observ. encoder | | [32] | | |
| | Prev. Action encoder* | | [16] | | |
| | Reward encoder* | | [16] | | |
| | RNN | | [128] | | |
| | MLP | [128,128,128] | | [256, 256] | |
| RL hparams | Learning rate | | 3e-4 | | |
| | Discount factor $\gamma$ | | 0.99 | | |
| | Smoothing coef $\tau$ | | 0.005 | | |
| | SAC temperature | | automatically updated by Haarnoja et al. (2018b) | | |
| | TD3 noises | | default values from Fujimoto et al. (2018) | | |
| | Replay buffer size | | 1e6 | | |
| | Batch size | 32 | | 64 | |
| Training hparams | Environment steps | 5M | 1.5M | 3M | |
| | Gradient steps | 0.1M | 1.5M | 0.6M | |
| Policy inputs | Largest input space | `oard` | `oar` | `oa` | `oar` |
| | Best input space | `oard` | `oa` | `o` | `o` |

Table 6: **Settings of the specialized methods we compared in the main paper.** For Meta-RL, we take the model on Cheetah-Vel as example.

| | Meta-RL* | POMDP | Robust RL | Gen. in RL |
|---|---|---|---|---|
| Approach | (off-policy) VariBAD | VRM | MRPO | EPOPT |
| Memory-based? | ✓ | ✓ | ✗ | ✗ |
| Off-policy? | ✓ | ✓ | ✗ | ✗ |
| Input space | `oard` | `oar` | `s` | `s` |
| Access to hidden states? | ✗ | ✗ | ✓ | ✓ |

et al., 2020) and VRM (Han et al., 2020). For MRPO (Jiang et al., 2021) and EPOPT (Rajeswaran et al., 2017a), they adopt totally different settings, i.e. on-policy Markovian approaches to MDPs (with access to the ground-truth state (`s`) of environment). Thus, in fact, MRPO and EPOPT should be more viewed as *oracle* policies as upper bounds of recurrent model-free RL.

## C EVALUATION DETAILS

The bar charts in Fig. 1 and 18 and Table 3 adopt the final performance of each method. We run each instance/variant in our method and each compared method with 4 random seeds. The final performance is calculated by the average performance of the last 20% environment steps across the 4 seeds.

In terms of selecting the best variant in our method for each subarea, we first compute the final performance of each variant, then normalize the final performance into $[0, 1]$, and finally select the best variant in the average of the normalized final performance across all the environments in each subarea.

For the bar charts in Fig. 1 and 18, we show the normalized returns of each method, calculated by $\frac{R - R_{\min}}{R_{\max} - R_{\min}} \in [0, 1]$, where $R$ is the raw average return of that method and $R_{\max}$ and $R_{\min}$ are the maximum and minimum of all the methods including Oracle policy and Random policy.

# D ENVIRONMENT DETAILS

## D.1 "STANDARD" POMDP

Except for the classic Pendulum environment, we use PyBullet (Coumans & Bai, 2016) as the simulator for "standard" POMDP environments. As the practice in VRM (Han et al., 2020), we remove all the position/angle-related entries in the observation space for "-V" environments and velocity-related entries for "-P" environments, to transform the original MDP into POMDP.

**{Pendulum,Ant,Cheetah,Hopper,Walker}-P.** The "-P" stands for the environments that keep position-related entries by removal of velocity-related entries. Thus, the observed state $s^o$ includes positions $p$, while the hidden state $s^h$ is the velocities $v$.

**{Pendulum,Ant,Cheetah,Hopper,Walker}-V.** The "-V" stands for the environments that keep velocity-related entries by removal of position-related entries. Thus, the observed state $s^o$ includes positions $v$, while the hidden state $s^h$ is the velocities $p$.

## D.2 META RL

**Semi-Circle.** We directly follow off-policy version of VariBAD (Dorfman et al., 2020). The observed state $s^o$ includes the agent's 2D position $p$, and the hidden state $s^h$ is referred to the goal state $p_g$. The goal state only appears in reward function: $R(s_t^o, s_{t+1}^o, a_t, s^h) \equiv R(p_{t+1}, p_g) = \mathbb{1}(\|p_{t+1} - p_g\|_2 \leq r)$. The dynamic function $T$ is independent of the goal state.

**Cheetah-Vel.** We directly follow Dorfman et al. (2020) using MuJoCo (Todorov et al., 2012) simulator of `HalfCheetah-v2`. The hidden state $s^h$ is the target velocity $v_g$ and observed state $s^o$ includes the velocity $v$. Reward function includes both the hidden state and action: $R(s_t^o, s_{t+1}^o, a_t, s^h) \equiv R(v_t, v_g, a_t) = -\|v_t - v_g\|_1 - 0.05\|a_t\|_2$. The dynamic function $T$ is independent of the goal state.

**Wind.** We modified the parameters of Wind environment in Dorfman et al. (2020) to make it harder to solve. The agent must navigate to a fixed (but unknown) goal $p_g$ within a distance of $D = 1$ from its fixed initial state. Similarly to Semi-Circle, the reward function is goal conditioned but without hidden state: $R(s_t^o, s_{t+1}^o, a_t, s^h) \equiv R(p_{t+1}, p_g) = \mathbb{1}(\|p_{t+1} - p_g\|_2 \leq r)$. The hidden state $s^h$ appear in the deterministic dynamics as a noise term, i.e. $s_{t+1}^o = s_t^o + a_t + s^h$, where $s^h$ is sampled from $U[-0.08, 0.08]$ at the initial time-step and then kept fixed.

## D.3 ROBUST RL

**{Hopper,Walker,Cheetah}-Robust.** We directly adopt the environments used in MRPO (Jiang et al., 2021). In each environment, the hidden state is the dynamics parameters including the density and friction coefficients of the simulated robot in roboschool, adapted from the SunBlaze (Packer et al., 2018). The exact ranges of the hidden states in each environment can be found in Table 1 of MRPO (Jiang et al., 2021). We evaluate the algorithms with 100 tasks in each environment, and use the average of them as average returns, and the average of the worst $10\%$ of them as worst returns, following MRPO paper.

## D.4 GENERALIZATION IN RL

**{Hopper,Cheetah}-Generalize.** We directly adopt the environments used in SunBlaze (Packer et al., 2018). In each environment, the hidden state is the dynamics parameters including the density, friction coefficients, and the power of the simulated robot in roboschool. The exact ranges of both interpolation and extrapolation in the hidden state distribution for each environment can be found in Table 1 of SunBlaze (Packer et al., 2018). We follow the practice of SunBlaze to evaluate interpolation and extrapolation success rates.

# E    FULL EXPERIMENTAL RESULTS

## E.1    LEARNING CURVES OF ALL THE COMPARED METHODS

In this subsection, we show all the learning curves of all the compared methods (including Oracle policy as upper bound, Markovian and Random policies as lower bounds) in each subarea, namely "Standard" POMDPs (Fig. 7 and Fig. 8), meta RL (Fig. 9), robust RL (Fig. 10), and generalization in RL (Fig. 11). The final performance of these learning curves generate the bar charts in Sec. 5.1.

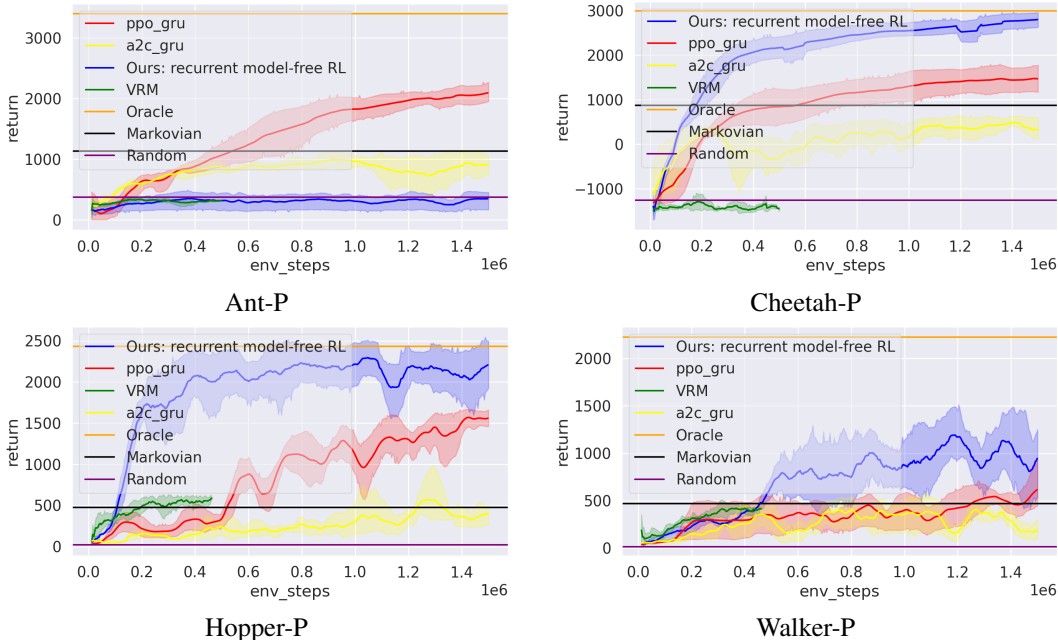

Figure 7: **Learning curves on "standard" POMDP environments that preserve positions & angles but occlude velocities in the states in PyBullet (Coumans & Bai, 2016) (namely "-P").** We show the results from the **single best variant** of our implementation on recurrent model-free RL, the popular recurrent model-free on-policy implementation (PPO-GRU, A2C-GRU) (Kostrikov, 2018), and also model-based method VRM (Han et al., 2020). Note that VRM is around 5x slower than ours, so we have to run 0.5M environment steps for it.

## E.2    SINGLE FACTOR ANALYSIS ON OUR METHOD

Our analysis will focus on ablating the important design decisions: the actor-critic architecture (`Arch`), the policy input space (`Inputs`), the underlying model-free RL algorithm (`RL`), the RNN encoder (`Encoder`), and the RNN context length (`Len`). As there are several decision variables in our method, we could only show the results of each variable in the plots by averaging the performance over the other variables, which we call **single factor analysis**. In this kind of analysis, we can only say one value is better than another in one factor in the average sense (not the maximal sense); therefore it can show the robustness of each factor, but cannot tell the best choices (showed in Sec. 5.1). We show single factor analysis plots in each subarea, namely "Standard" POMDPs (Fig. 12 and Fig. 13), meta RL (Fig. 14), robust RL (Fig. 15), and generalization in RL (Fig. 16).

From these plots, we can see that each decision factor can make a difference in some environments. For example, the choice of RL algorithm is crucial in Ant-P (Fig. 12), Cheetah-V (Fig. 13), Wind (Fig. 14) and Hopper-Generalize (Fig. 16). The context length is essential in all the "-P" environments (Fig. 12), Cheetah-Vel (Fig. 14), and both the generalization environments (Fig. 16). The policy input space can make a difference in most "-P" environments (Fig. 12) possibly because `oar` contains the information of missing velocities.

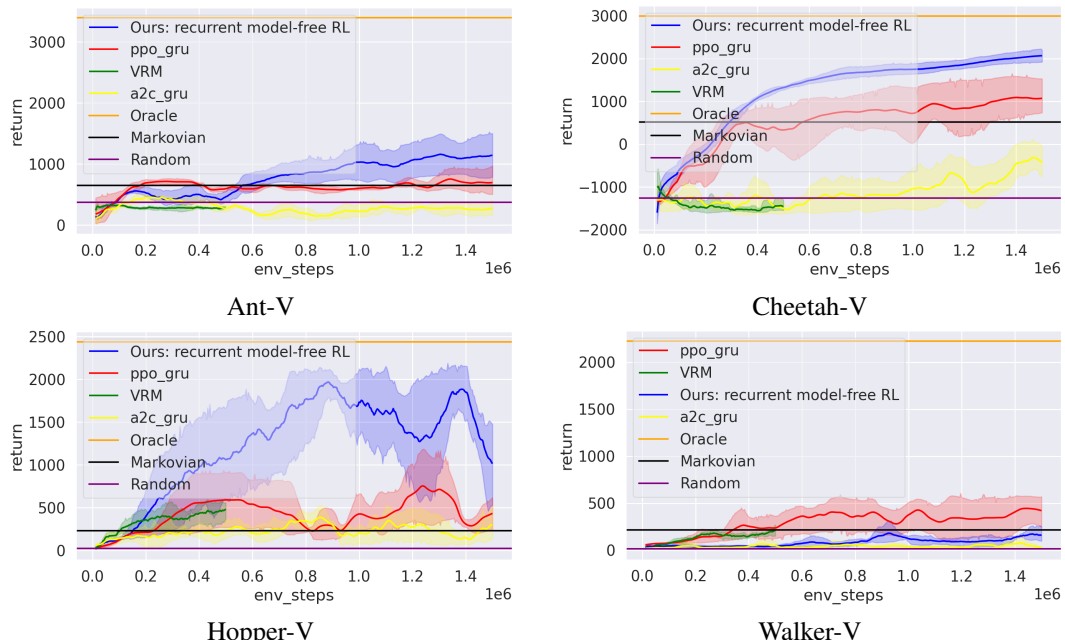

Figure 8: **Learning curves on "standard" POMDP environments that preserve velocities but occlude positions & angles in the states in PyBullet (Coumans & Bai, 2016) (namely "-V").** We show the results from the **single best variant** of our implementation on recurrent model-free RL, the popular recurrent model-free on-policy implementation (PPO-GRU, A2C-GRU) (Kostrikov, 2018), and also model-based method VRM (Han et al., 2020). Note that VRM is around 5x slower than ours, so we have to run 0.5M environment steps for it.

### E.3 ADDITIONAL RESULTS ON SEPARATE VS SHARED RECURRENT ACTOR-CRITIC ARCHITECTURE

In Fig. 5 of Sec. 5.2, we show the importance of selecting a separate recurrent actor-critic architecture in a meta RL environment, Semi-Circle. Now we show the result in another POMDP environment, Pendulum-V, which occludes the positions and angles, in Fig. 17. We can see that the shared encoder architecture is also worse than the separate one, possibly due to the different gradient scales in actor and critic losses w.r.t. the encoder.

### E.4 ADDITIONAL RESULTS ON COMPARISON WITH VRM

Both Fig. 1 and Fig. 18 shows the final performance of the same single variant of our implementation, but the former shows our results with 1.5M simulation steps while the latter shows our results with 0.5M simulation steps to match with those of VRM due to the time budget.

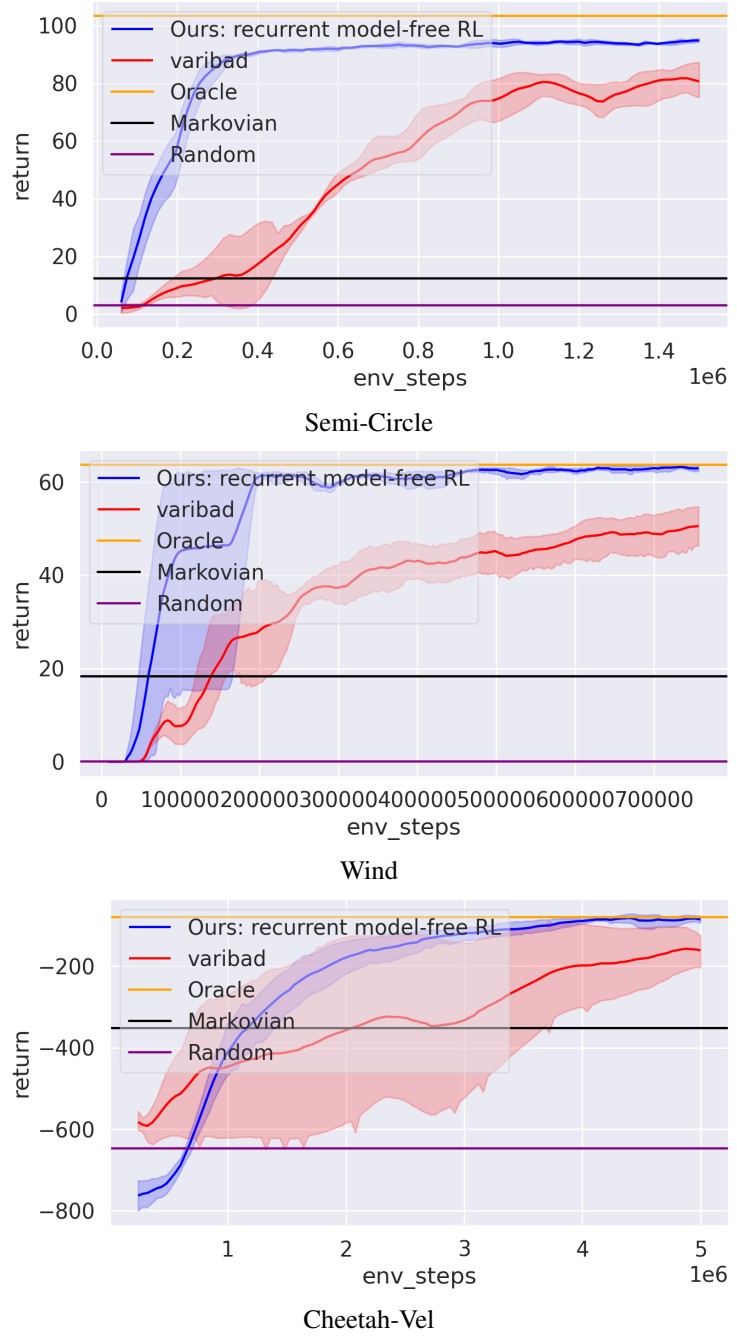

Figure 9: **Learning curves on meta RL environments.** We show the results from the **single best variant** of our implementation on recurrent model-free RL, the specialized meta RL method VariBAD (Dorfman et al., 2020)

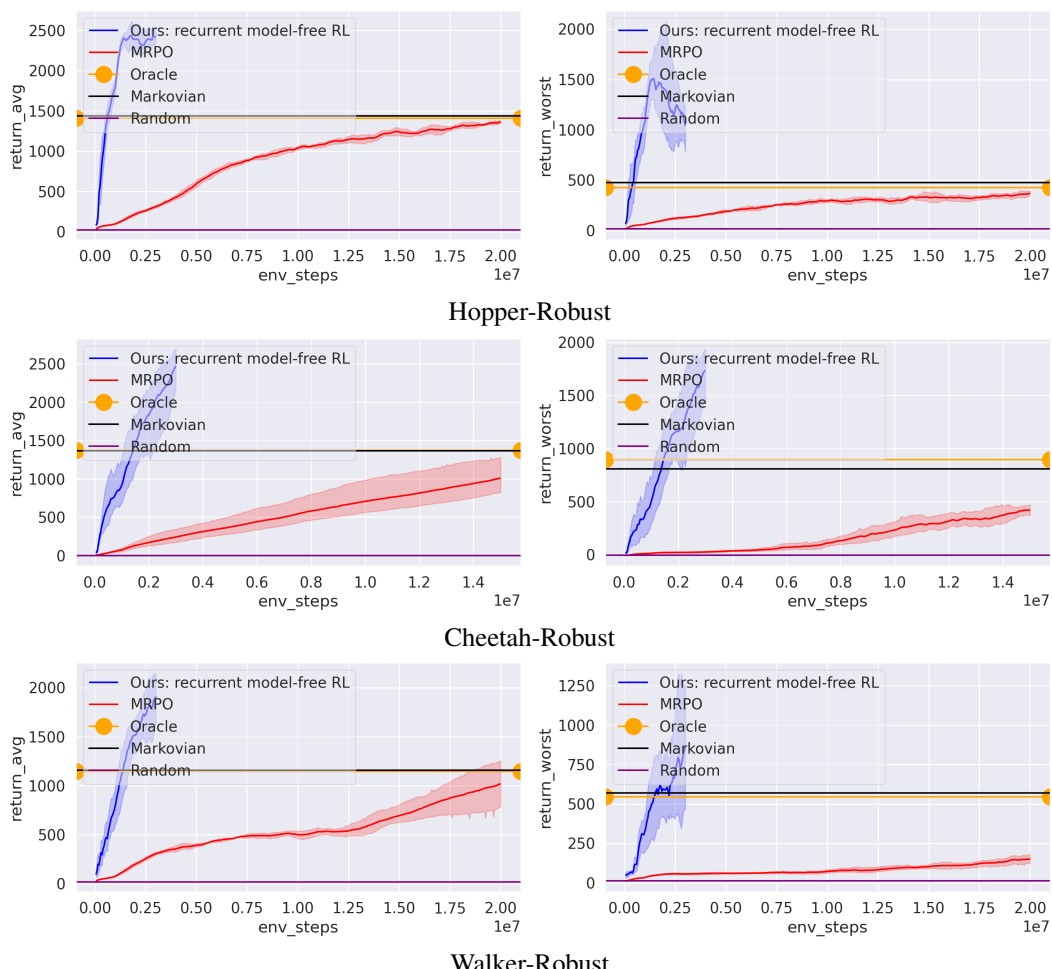

Figure 10: **Learning curves on robust RL environments.** We show the average returns (left figures) and worst returns (right figures) from the **single best variant** of our implementation on recurrent model-free RL, the specialized robust RL method MRPO (Jiang et al., 2021). Note that our method is much slower than MRPO, so we have to run our method within 3M environment steps. But the results show that our method have much better sample efficiency over MRPO.

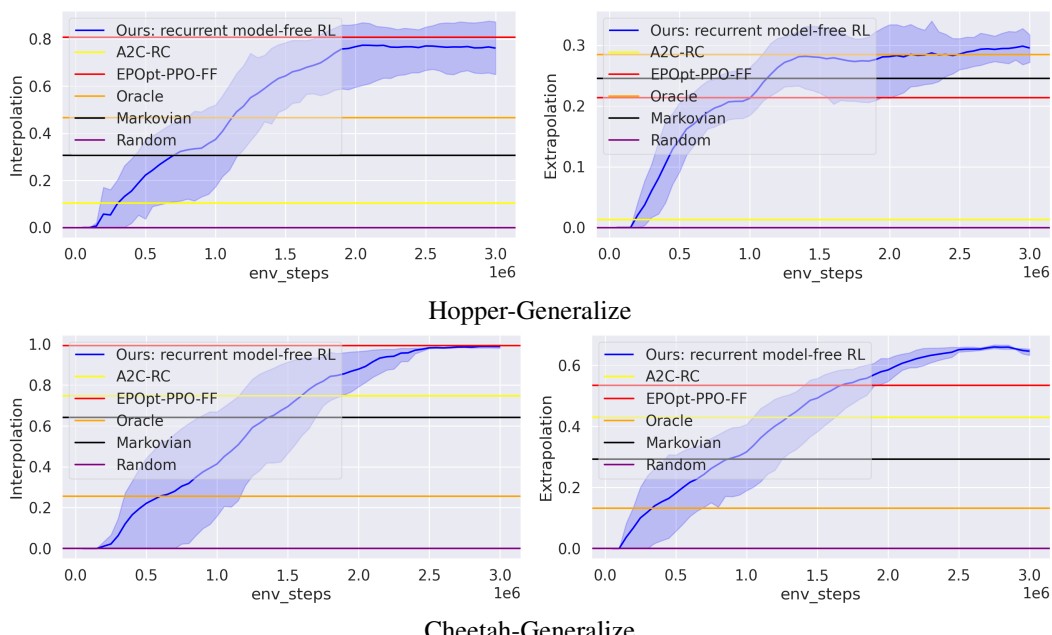

Hopper-Generalize

Cheetah-Generalize

Figure 11: **Learning curves on generalization in RL environments.** We show the interpolation success rates (left figures) and extrapolation success rates (right figures) from the **single best variant** of our implementation on recurrent model-free RL. We also show the final performance of the specialized method EPOpt-PPO-FF (Rajeswaran et al., 2017a) and another recurrent model-free (on-policy) RL method (A2C-RC) copied from the Table 7 & 8 in Packer et al. (2018).

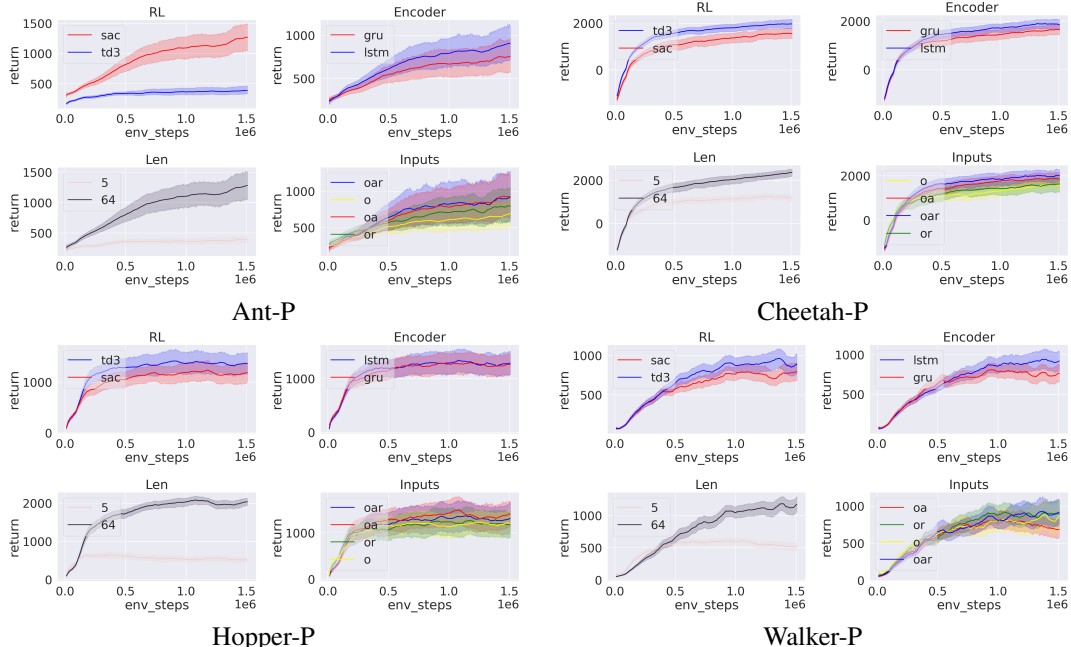

Figure 12: **Ablation study of our implementation on "standard" POMDP environments that preserve positions & angles but occlude velocities in the states in pybullet (Coumans & Bai, 2016) (namely "-P").** We show the single factor analysis on the 4 decision factors including RL, Encoder, Len, and Inputs for each environment.

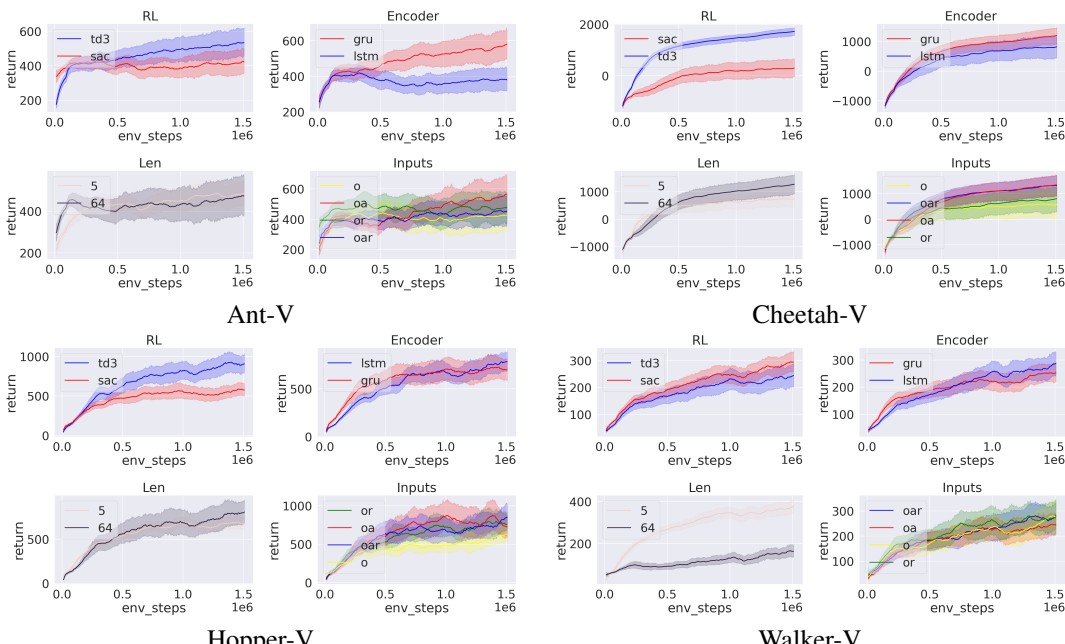

Figure 13: **Ablation study of our implementation on "standard" POMDP environments that preserve velocities but occlude positions & angles in the states in pybullet (**Coumans & Bai**, 2016) (namely "-V").** We show the single factor analysis on the 4 decision factors including RL, Encoder, Len, and Inputs for each environment.

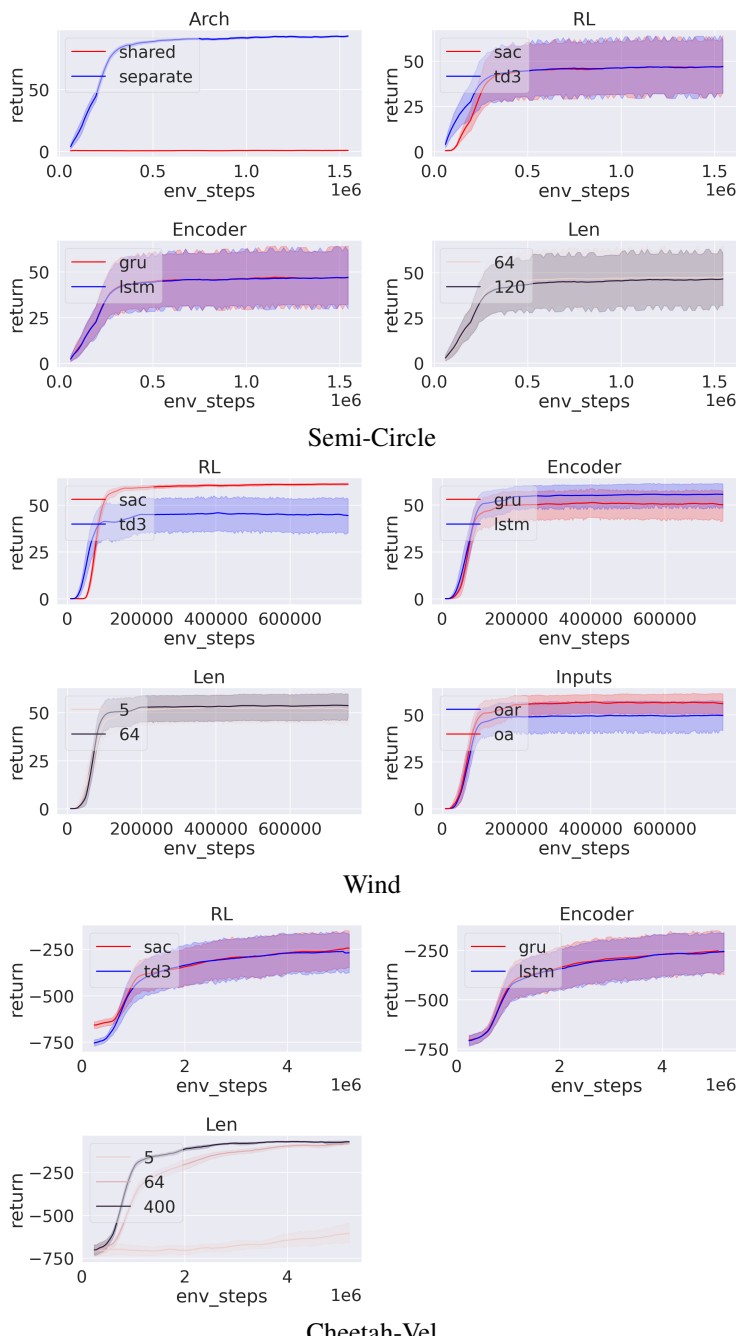

Figure 14: **Ablation study of our implementation on meta RL environments.** We show the single factor analysis on covering all the decision factors

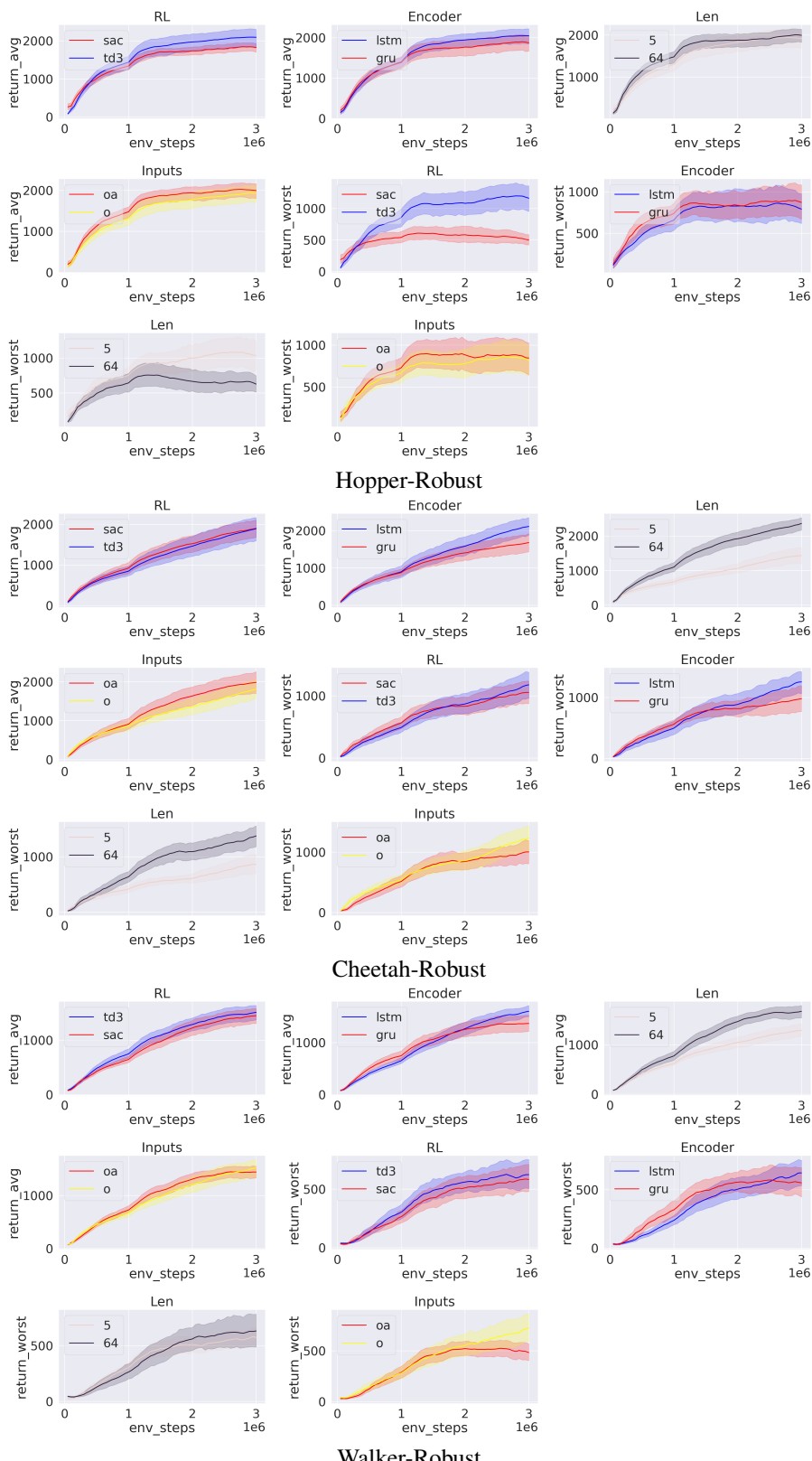

Figure 15: **Ablation study of our implementation on robust RL environments.** We show the single factor analysis on the 4 decision factors including RL, Encoder, Len, and Inputs for each environment in both average returns and worst returns.

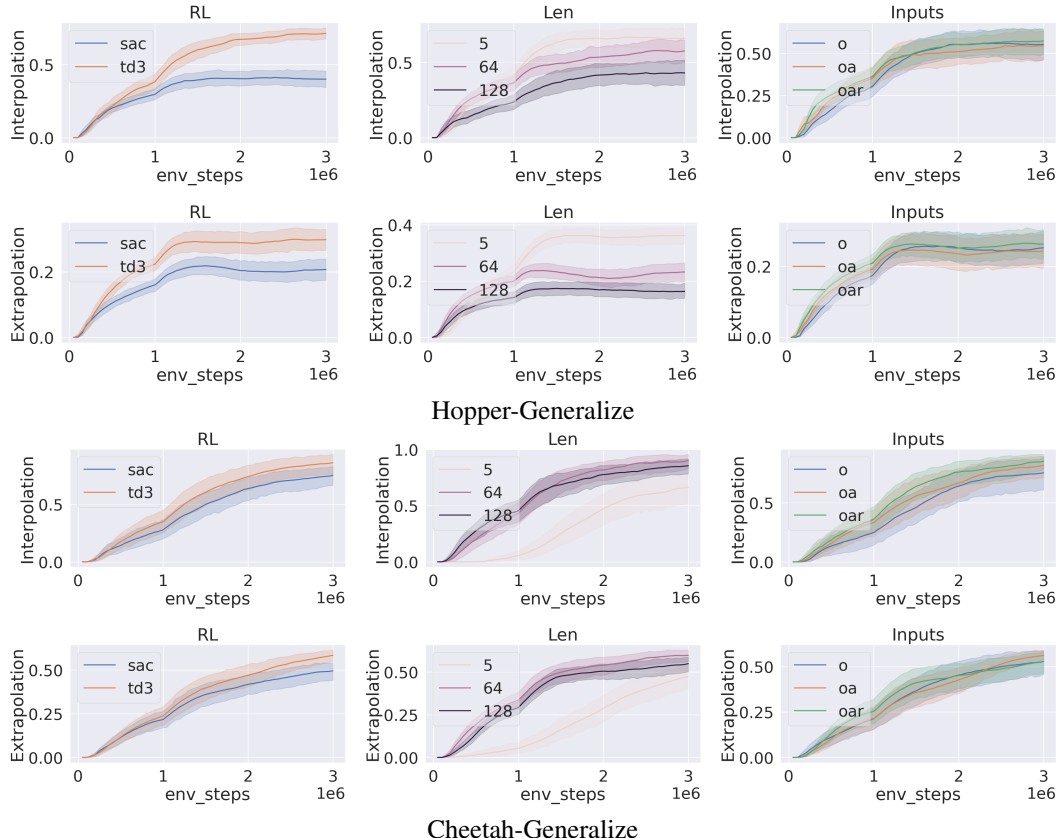

Figure 16: **Ablation study of our implementation on generalization in RL environments.** We show the single factor analysis on the 3 decision factors including RL, Len, and Inputs for each environment in both interpolation and extraploation success rates.

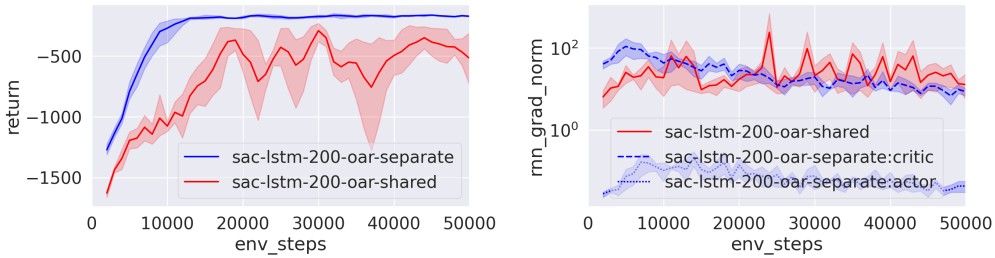

Figure 17: **Comparison between** *shared* **and** *separate* **recurrent actor-critic architecture** with all the other hyperparameters same, on Pendulum-V, a toy "standard" POMDP environment. We show the performance metric (left) and also the gradient norm of the RNN encoder(s) (right, in **log-scale**). For the separate architecture, `:critic` and `:actor` refer to the separate RNN in critic and actor networks, respectively.

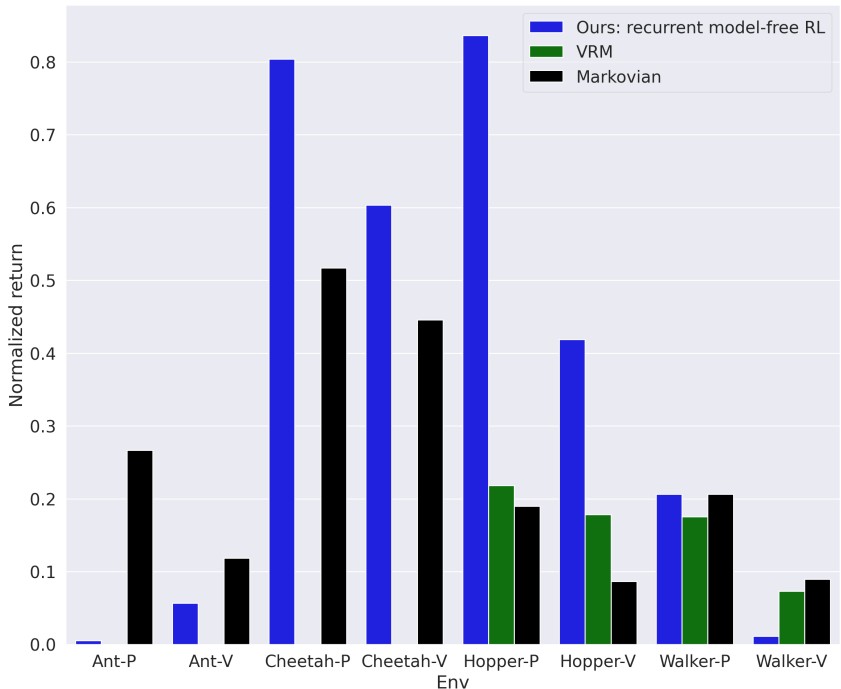

Figure 18: Final **normalized returns** of our implemented recurrent model-free RL algorithm with the same hyperparameters, and the prior method VRM (Han et al., 2020) across the eight environments in **"standard" POMDPs**, each of which trained in 0.5M simulation steps.

# F    ABLATION STUDY ON RNN ARCHITECTURES

Our implementation of recurrent model-free RL uses the popular 1-layer LSTM or GRU. To investigate the effect of RNN architecture, we ablate two RNN variants: one is 2-layer stacked LSTM/GRU, the other is Particle-Filter RNN (PF-RNN (Ma et al., 2020a)). PF-RNN maintains a stochastic belief (posterior distribution) through a set of weighted particles to better capture the uncertainty and multi-modality, compared to classic RNN's deterministic belief. DPFRL (Ma et al., 2020b) applies PF-GRU to POMDP tasks, using the mean of particles and MGF features of particles as the belief state. We follow the implementation of PF-GRU in DPFRL (Ma et al., 2020b) to replace regular GRU, keeping the other model components the same.

We try 2-layer LSTM/GRU, on our best variant across all the environments. We run PF-GRU on "standard" POMDP environments where the best variant also uses GRU for fair comparison, and do not try PF-LSTM as it is not adopted in DPFRL (Ma et al., 2020b).

Since 2-layer LSTM/GRU doubles the training time of 1-layer LSTM/GRU given the same gradient updating frequency, we have to decrease the frequency from 1.0 to 0.6 in "standard" POMDPs. Similarly, PF-GRU costs $15\times$ than 1-layer GRU, so we also have to decrease the frequency from 1.0 to 0.6. All the other hyperparameters remain the same.

Figs. 19 to 23 show all the learning curves of the ablation study on RNN architectures. Except for rare cases (e.g. Ant-P, Walker-V), the RNN variants perform worse than or are par with 1-layer LSTM/GRU, although in most cases the RNN variants can still outperform the Markovian policies. Possible reasons are that the RNN variants need hyperparameter tuning and more training samples to converge.

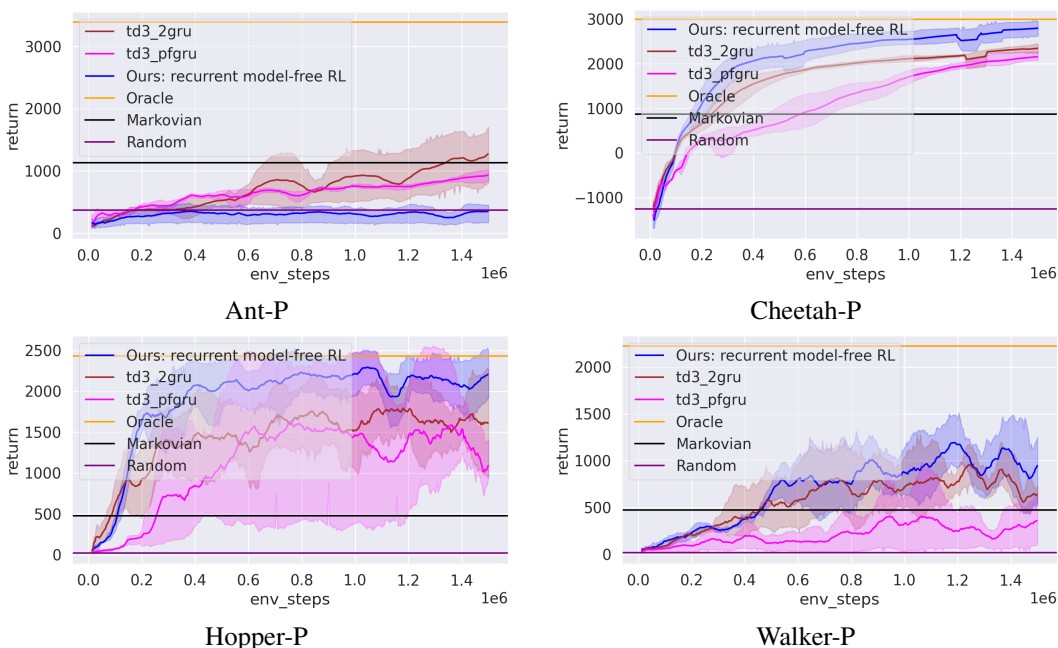

Figure 19: **Ablation study on RNN architecture in "standard" POMDPs ("-P").** We show two models that are only different from our best single variant in the RNN architectures, namely using **2-layer stacked GRU** (td3-2gru) and **PF-GRU** (Ma et al., 2020a;b) (td3-pfgru), instead of 1-layer GRU.

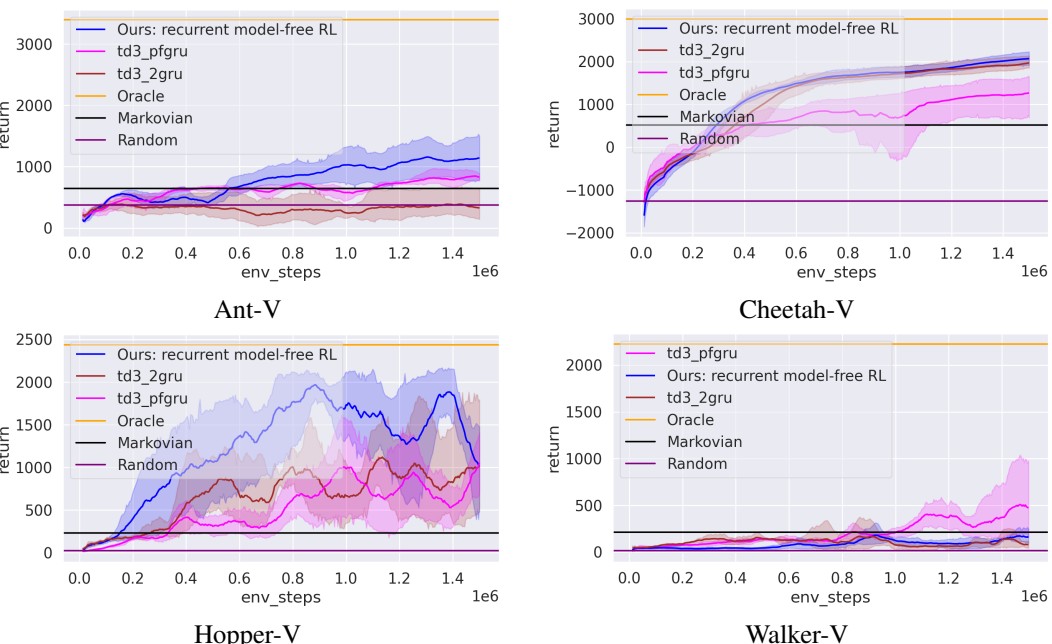

Figure 20: **Ablation study on RNN architecture in "standard" POMDPs ("-V").** We show two models that are only different from our best single variant in the RNN architectures, namely using **2-layer stacked GRU** (td3-2gru) and **PF-GRU** (Ma et al., 2020a;b) (td3-pfgru), instead of 1-layer GRU.

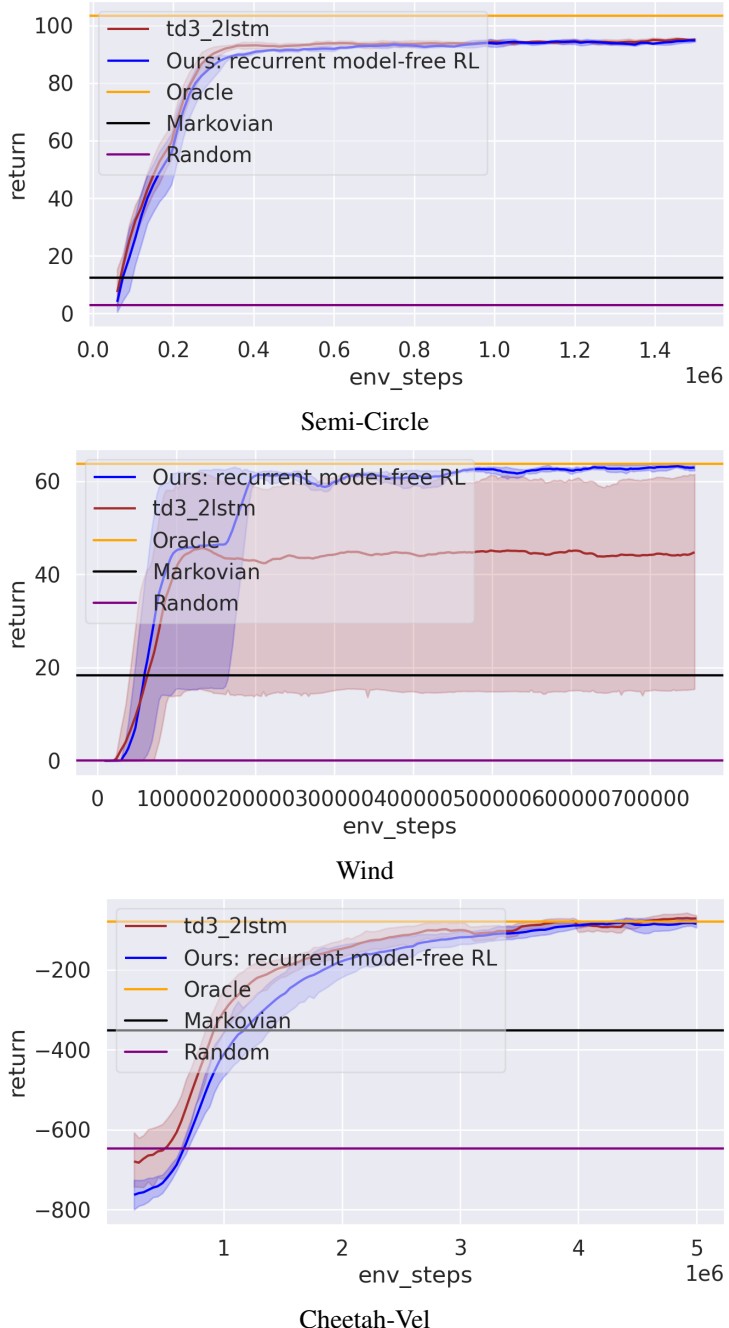

Figure 21: **Ablation study on RNN architecture in Meta RL.** We show one model that are only different from our best single variant in the RNN architecture, namely using **2-layer stacked LSTM** (td3-2lstm), instead of 1-layer LSTM.

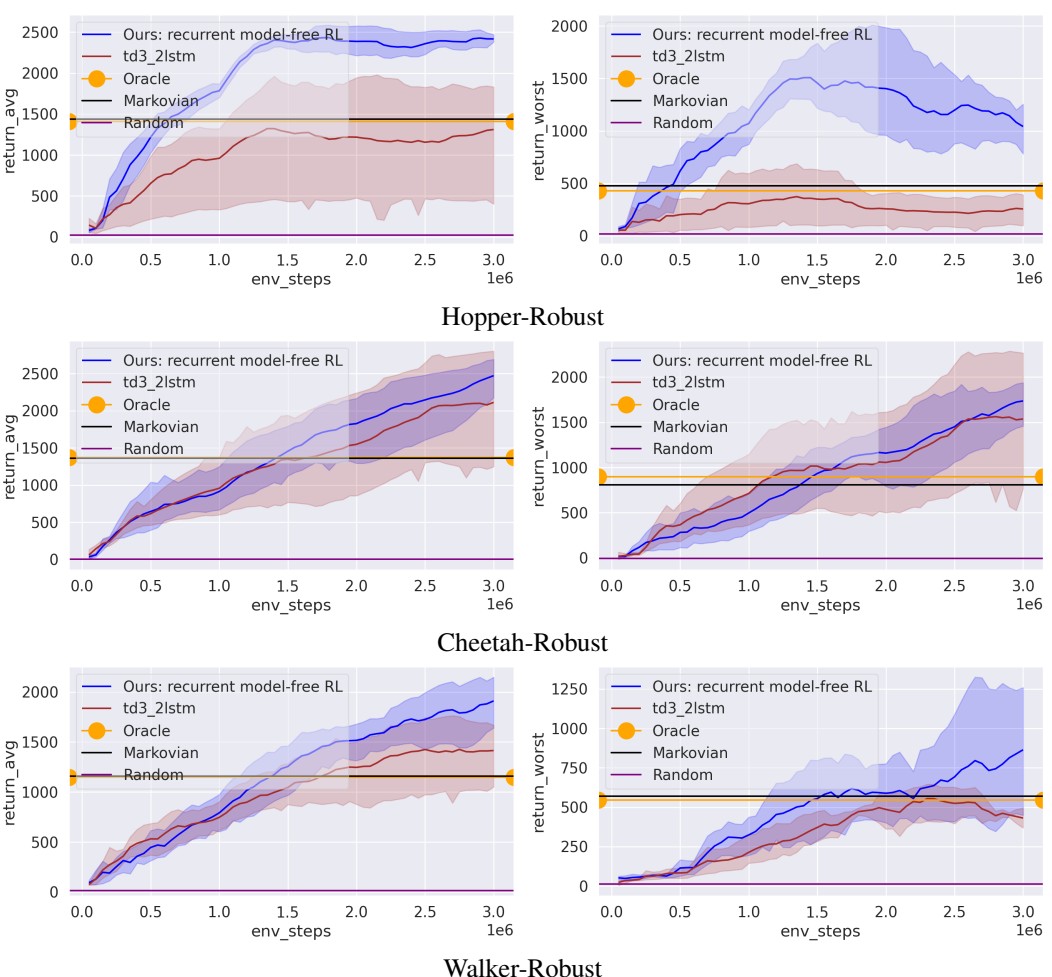

Figure 22: **Ablation study on RNN architecture in robust RL.** We show one model that are only different from our best single variant in the RNN architecture, namely using **2-layer stacked LSTM** (td3-2lstm), instead of 1-layer LSTM.

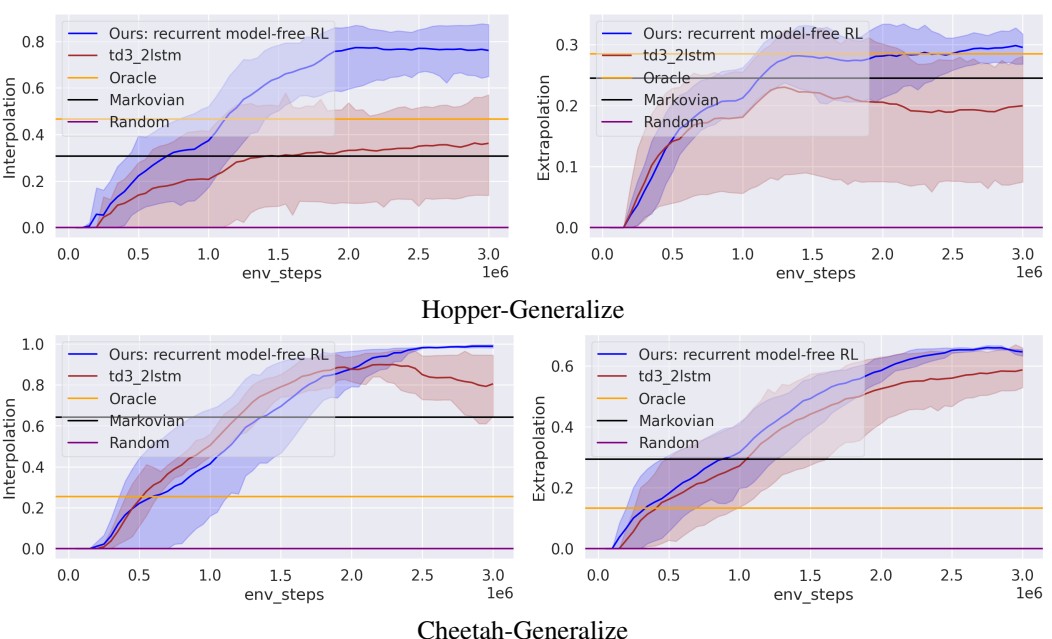

Figure 23: **Ablation study on RNN architecture in generalization in RL.** We show one model that are only different from our best single variant in the RNN architecture, namely using **2-layer stacked LSTM** (td3-2lstm), instead of 1-layer LSTM.

# G  ADDITIONAL RESULTS ON COMPARISON WITH OFF-POLICY VARIBAD

In our experiment section (Sec. 5) and Fig. 9, we show the learning curves of our method and our *re-implemented* off-policy VariBAD (Dorfman et al., 2020). To rule out the possibility of re-implementation bugs in the off-policy VariBAD, we ran the official off-policy VariBAD implementation[2], denoted as **VariBAD-BOReL**.

In Fig. 24, we show official off-policy VariBAD (VariBAD-BOReL in short), our re-implemented off-policy VariBAD (VariBAD in short), and one variant of our recurrent model-free RL, which uses the same decision factors as VariBAD-BOReL (SAC as RL algorithm, GRU as encoder, `oar` as input, 400 as context length) for fair comparison. We also try Ant-Dir from Dorfman et al. (2020) paper, a more challenging meta RL environment than Cheetah-Vel.

In Semi-Circle and Wind, VariBAD-BOReL has similar performance as our re-implemented version (VariBAD).

In Cheetah-Vel, we find VariBAD-BOReL outperforms our re-implemented version, but still has a gap from what they reported in Fig. 11 in the appendix of Dorfman et al. (2020). Our implementation still outperforms VariBAD-BOReL, supporting our claim in the main paper.

Finally, in the newly added and more challenging **Ant-Dir**, we found our recurrent model-free RL can even greatly surpass Oracle (trained with same environment and gradient steps), while VariBAD-BOReL somehow has extremely low performance (worse than Random) and suffers from numerical issue to terminate early.

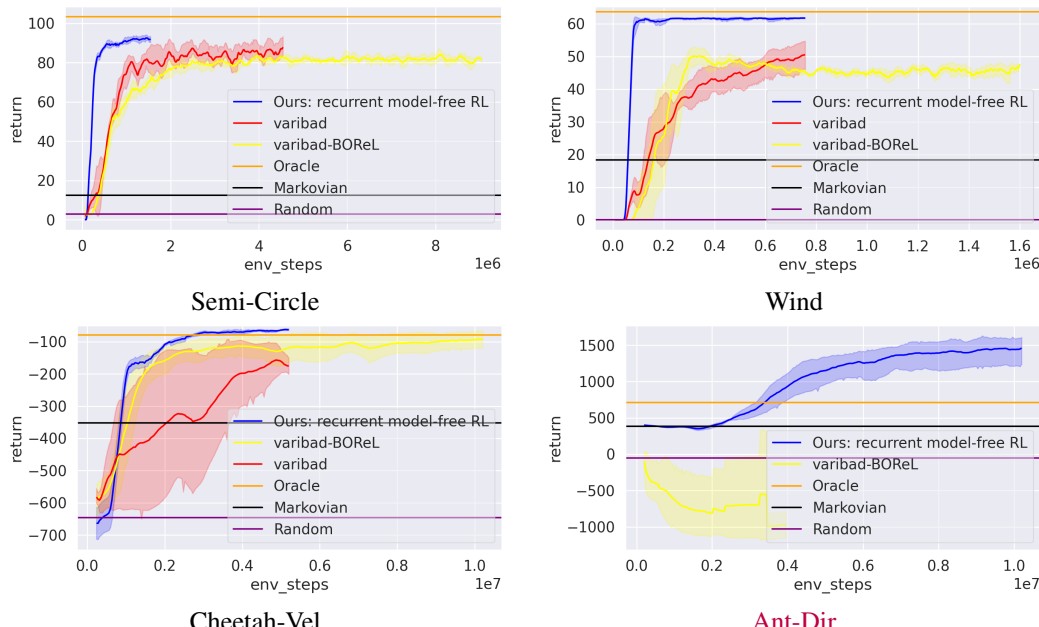

Figure 24: **Additional learning curves on meta RL environments.** We show the results from the **single variant** of our implementation on recurrent model-free RL (sac-gru-oar-400), and the specialized meta RL method off-policy VariBAD (Dorfman et al., 2020) (their official implementation **VariBAD-BOReL** and our re-implementation **VariBAD**).

---

[2]https://github.com/Rondorf/BOReL

