# OpenReview forum: "Recurrent Model-Free RL is a Strong Baseline for Many POMDPs"
_ICLR.cc/2022/Conference — ICLR 2022 Submitted_

### Official Review · Reviewer_2WFY · 2021-10-24

**Correctness:** 2
**Technical Novelty And Significance:** 3
**Empirical Novelty And Significance:** 3
**Recommendation:** 5
**Confidence:** 4

**Main Review:**

This work is quite straightforward and simple - which is good. The authors state that an end-to-end approach can be competitive with sota methods on tailored problems, if only specific considerations are taken into account.

The main drawback of this work is that the results are inconclusive.
1. There is no theoretical reasoning behind the authors claims. No motivating principles. It is hard for me to assess the correctness of their claims solely from their experiments on 4 mujoco tasks.
2. While the authors did run extensive experiments on the 4 mujoco tasks, these are definitely not enough. For the claims of this paper, I would at least want to see extensive experiments on another large domain (e.g., 4 other environments in atari, and perhaps at least 1 hard task in another domain).
3. When do things fail? Results that only show successes usually feel cherry picked. I assume the authors can find environments and/or algorithms for which their design choices fail. It is important to show these as well to understand the tradeoffs better.

**Summary Of The Paper:**

This paper is concerned with understanding when RNNs are useful in POMDPs. They should that, for a class of POMDPS (e.g., meta RL, robust RL), standard RNNs can be competitive with solutions that are tailored to the given POMDP structure. The authors discuss four design considerations that they claim are essential for performance: (1) decoupling actor and critic networks, (2) using off-policy instead of on-policy algorithms, (3) context length of RNN, and (4) using rewards as historical input. They should on 4 mujoco baselines that their results are competitive with sota methods whenever these design choices are taken into account.

**Summary Of The Review:**

Good direction, results are inconclusive.

---

> ### Author Response · Authors · 2021-11-20
> **Authors’ Response**
>
> We thank the reviewer for their insightful suggestions. The reviewer's main concerns seem to be about the number of environments used and a lack of theoretical analysis. The concern about the number of environments is likely a simple misunderstanding: we used a total of 16 environments in our experiments (not 4). While we agree that additional theoretical analysis would strengthen the paper, we believe that our empirical analysis represents a solid empirical contribution; ICLR has a long history of recognizing papers that have only empirical contributions [1-4]. We have also revised the ablation study section (Sec 5.2) to address the comments.
>
> **Does the reviewer have specific, additional ablation experiments or failure cases they would like to see examined?**
>
> [1] Logan Engstrom, Andrew Ilyas, Shibani Santurkar, Dimitris Tsipras, Firdaus Janoos, Larry Rudolph, and Alek- sander Madry. Implementation matters in deep RL: A case study on PPO and TRPO. In 8th International Conference on Learning Representations, ICLR 2020.
>
> [2] Marcin Andrychowicz, Anton Raichuk, Piotr Stanczyk, Manu Orsini, Sertan Girgin, Raphae ̈l Marinier, Le ́onard Hussenot, Matthieu Geist, Olivier Pietquin, Marcin Michalski, Sylvain Gelly, and Olivier Bachem. What matters for on-policy deep actor-critic methods? A large-scale study. In 9th International Conference on Learning Representations, ICLR 2021.
>
> [3] Mariya Toneva and Alessandro Sordoni and Remi Tachet des Combes and Adam Trischler and Yoshua Bengio and Geoffrey J. Gordon. An Empirical Study of Example Forgetting during Deep Neural Network Learning. ICLR 2019.
>
> [4] Roman Novak and Yasaman Bahri and Daniel A. Abolafia and Jeffrey Pennington and Jascha Sohl-Dickstein. Sensitivity and Generalization in Neural Networks: an Empirical Study. ICLR 2018.
>
> > Inconclusive results. They only show 4 mujoco tasks, not enough
>
> We believe there is some misunderstanding on our experiment environments. We ran a total of 16 different environments (not 4):
> - Standard POMDPs (8): 4 PyBullet environments x 2 tasks = 8 tasks
> - Meta RL (3) : 2 continuous gridworlds, one of them with sparse reward (Semi-Circle), and 1 MuJoCo environment
> - Robust RL (3): 3 RoboSchool environments
> - Generalization in RL (2): 2 RoboSchool environments
>
> We have already included 16 tasks, which is substantially larger than most prior work. For example, VariBAD [5] uses 4 MuJoCo tasks and 1 gridworld, VRM [6] uses 6 roboschool tasks, 4 classic robotics tasks and 1 memorization task. MRPO [7] uses 3 roboschool tasks and 3 classic robotics tasks.
>
> [5] Zintgraf, Luisa, et al. "Varibad: A very good method for bayes-adaptive deep rl via meta-learning." arXiv preprint arXiv:1910.08348 (2019).
>
> [6] Han, Dongqi, Kenji Doya, and Jun Tani. "Variational recurrent models for solving partially observable control tasks." arXiv preprint arXiv:1912.10703 (2019).
>
> [7] Jiang, Yuankun, et al. "Monotonic robust policy optimization with model discrepancy." International Conference on Machine Learning. PMLR, 2021.
>
> > Want to see extensive experiments on another large domain (such as atari)
>
> Our experiments already cover 16 tasks, ranging from simple (1-dimensional observations) to more complex (26-dimensional observations),  where even the best method achieves less than 30% success (extrapolation in Hopper-Generalize). These environments already cover 4 domains: generalization in RL, robustness in RL, meta RL, and more conventional POMDPs. We think that 16 tasks are sufficient for substantiating our claims, and goes well beyond the standard set in prior work.
>
> We will investigate the feasibility of comparing to Atari, but note that this will take much longer than the rebuttal period.
>
> > No theory reasoning, no motivating principles
>
> As an empirical paper, we do not aim to have theoretical contributions.  For the motivating principle, most of our design decisions are unanimous to tasks. We have some intuition on the choice of policy input and context length. For example, in Meta RL environments (Cheetah-Vel and Semi-Circle) where the hidden states only appears in rewards, it is important to add rewards into policy inputs. In sparse reward setting of Meta RL environments (Semi-Circle), using a relatively long context length (such as 64) is necessary to infer the hidden states.
>
> In the new version of our paper, we provide more analysis in our ablation study in Sec 5.2.
>
> > When do things fail? Results that only show successes usually feel cherry-picked. Authors can find environments and/or algorithms for which their design choices fail.
>
> Yes, there are environments where our best design choices fail. For example, Table 3 shows several failure cases of our best single variants and how one decision factor change can improve the performance. We also provide some explanation for failure in Sec. 5.2.

---

> > ### Author Response · Authors · 2021-11-22
> > **Add a challenging environment**
> >
> > > Perhaps at least 1 hard task in another domain
> >
> > As suggested by the reviewer, we have added one challenging meta-RL environment, Ant-Dir, in our latest version of paper. Please see appendix G for details. We found our recurrent model-free RL can perform well in Ant-Dir, much better than Oracle and VariBAD-BOReL. So we believe the promise of recurrent model-free RL in these challenging domains. **Do these new experiments address the reviewer's concerns about the experiments?**

---

> > ### Comment · Reviewer_2WFY · 2021-11-26
> > **Response to Rebuttal**
> >
> > I thank the authors for their response. I still believe this work, while touching on an important subject, does not provide sufficient evidence for its claims.
> >
> > While the authors did run a substantial amount of experiments on standard continuous control benchmarks, there are several factors that make these experiments not satisfying. First, there are many factors that these environments don't take in account, including: sparse rewards, longer horizon, discrete action space, etc.
> >
> > If the authors wish to provide purely empirical evidence, it must hold for a wide variety of environments, e.g., all of the atari suite which provides a wide spectrum of challenges in RL.
> >
> > Regarding theory. As long as the authors wish to focus on very specific types of tasks (e.g., arguably "easy" continuous control benchmarks), they must find a way to argue their results will hold more broadly. Theory is one way to do this. Otherwise, the authors should provide extensive experiments on a variety of domains.
> >
> > I would like to emphasize that this paper should be carefully scrutinized due to its significant claims, which (if they indeed hold more broadly) could have significant impact on RL research. This should not go about lightly. If these claims do not hold more broadly, this paper may have the exact opposite effect on RL research.

---

> > > ### Author Response · Authors · 2021-12-01
> > > **Re: Response**
> > >
> > > Dear Reviewer,
> > >
> > > Thank you for clarifying the concerns. While we believe that the current claims in the paper are already quite narrow (e.g., we don't make any claims about sparse rewards or long horizons, only about performance on standard benchmarks) and supported by substantial empirical evidence (17 environments and a total of 6 baselines), we would be happy to soften any of claims in the paper.
> > >
> > > Based on this discussion, we will revise the paper to say that recurrent model-free RL outperforms *some* prior specialized methods on the *benchmarks* they used, not *all* prior specialized methods (Sec. 1 and 5).
> > >
> > > **Are there additional, specific claims in the paper that should be further narrowed**?
> > >
> > > Authors

---

### Official Review · Reviewer_3HH1 · 2021-10-30

**Correctness:** 3
**Technical Novelty And Significance:** 1
**Empirical Novelty And Significance:** 4
**Recommendation:** 8
**Confidence:** 4

**Main Review:**

Strengths:
  - A few grammatical mistakes aside, the paper is well-organized and understandable. It sets out a clear picture of the current POMDP landscape, making clear the distinctions between different classes of problem (such as stationary vs nonstationary, across episode, policy inputs) and the concerns that each raise. It describes in detail the process by which the experiments were chosen and performed.
  - It's surprising to me how novel the contribution is. This is not because any new methods are proposed, but rather because so little previous work has really considered the question of actually getting recurrent algorithms to work well. I think an analysis like this is long overdue, even if I may have some concerns over the experiments themselves.
  - A good amount of effort clearly went into reproducibility and efficient, readable code

Weaknesses:
  - I'm not as convinced as I'd like to be by the experiments, which are ultimately what this empirical study rest on. I'll detail a few issues:
    1) In each class of problem, the tuned RNN is only compared against one other method. In general, these appear to be recent, SOTA, and well-chosen, but between that and the use of bar graphs instead of learning curves, it's hard to really sanity check. How would something like PFRNNs or DRBPNs compare?
    2) As a follow-on, I'm not sure that these are really fair comparisons. For the comparisons to other methods, the authors use an off-policy algorithm (SAC) with access to information that is not necessarily common in POMDP literature (rewards and done signals aren't often counted in the stored history). VRM follows a relatively similar training procedure, as does the modified VariBAD, but the other two algorithms are quite different. Can these methods really be compared in terms of simulation steps?
    3) While the set of all possible RNN architecture choices is enormous, I do think that there were a few more that could have been considered. Most importantly, I'm not fully convinced by the shared vs separate argument. The hypothesis of exploding gradient should be relatively simple to empirically log and demonstrate, and it also heavily depends on the shape of the environment rewards. Most of these environments have large, dense reward signals which need normalization. Another consideration would be whether the RNN should be before or after feature extraction. Finally (though this is more something that I've come across in my own work), layer-normalized RNNs often vastly outperform their standard counterparts.
    4) I preferred the learning curves found in the appendix to the bar chart comparisons in the main body. I think it would be more informative to compare the methods with some error bars and a better idea of the process. Also, it seemed like some environments (like Cheetah-Vel) didn't learn? A couple baselines (optimal upper-bound, random policy lower-bound) would give a better idea of how well the methods are actually doing.

**Summary Of The Paper:**

This paper primarily provides an empirical analysis of recurrent model-free RL on several classes of POMDP, showing that if parameters are well-chosen, the basic approach of just applying a recurrent layer is not only competitive with but often outperforms specially-designed methods for those problem classes. The authors consider different architectures, algorithms, types of RNN, and other parameters and fill in a significant gap left by previous papers, which didn't consider most of the important details of getting model-free RNN methods to work well

**Summary Of The Review:**

I'm recommending this paper with some reservations. Overall, I really think work like this is valuable and illustrates a large gap in knowledge that was previously unaddressed. I've always found it frustrating that the approaches to partial observability are so ad-hoc. I also appreciate the clarity with which this paper addresses the classes of POMDP and the considerations required for each. That said, I have some concerns with the thoroughness of the experiments and would like to see more baselines and fairer comparisons.

---

> ### Author Response · Authors · 2021-11-20
> **Authors’ Response**
>
> We thank the reviewer for their insightful suggestions. The review mainly focuses on more baselines and RNN architectures, and also questions on fair comparison with specialized methods. In response to the concern, we have added the experiment results of PF-RNNs and Oracle baselines in the new version of our paper. We believe that the experiments in the paper are fair: all methods are compared using the same metric (taken from prior work) under the same assumptions (taken from prior work). We agree that some specialized methods may not have been designed with this metric (sample efficiency) in mind, and have revised the paper to note this.
>
> > A few grammatical mistakes aside, the paper is well-organized and understandable.
>
> Thank the reviewer for reminding us of the grammatical mistakes. We have fixed them in the new version.
>
> > Show the Learning curves in the main paper
>
> Thank you for the suggestion. In the new version, we have replaced the bar charts with learning curves (Fig 2,3,4).
>
> > it seemed like some environments (like Cheetah-Vel) didn't learn?
>
> Sorry for the confusion. Actually, our recurrent model-free RL learns well in Cheetah-Vel and is on par with Oracle, see Fig. 2. The reward is negative in Cheetah-Vel which causes confusion.
>
> > Add baselines (optimal upper bound, random policy lower bound)
>
> We have added both baselines, namely **Oracle** as optimal upper bound, **Random** as a lower bound in all the learning curves. The results of Oracle policies are from maximum of TD3 and SAC performance per task, trained with access to the hidden states (thus the POMDP becomes MDP), and with same hyperparameters as our recurrent RL.
>
> We found there is a gap between our best variant and Oracle in most tasks, which is understandable. In some tasks such as generalization in RL and robust RL, we found given same environment and gradient step updates, Oracle performs worse than our best variant of recurrent RL. We hypothesize that this is caused by some implicit regularization performed by recurrent architectures; we leave further investigation to future work.
>
> > Fair comparison between the specialized methods, in terms of simulation steps
>
> We believe that the comparisons in the paper are fair: all methods are compared using the same metric (return vs environment steps) under the same assumptions. Table 6 in Appendix B clarifies the assumptions of all methods. Our paper builds on prior work on POMDPs that focuses on sample efficiency [1] and that acknowledge that agent’s past history includes observations, actions, rewards, and terminal flags [1,2].
>
> [1] Dorfman, Ron, Idan Shenfeld, and Aviv Tamar. "Offline Meta Learning of Exploration." arXiv e-prints (2020).
>
> [2] Duan, Yan, et al. "Rl^2: Fast reinforcement learning via slow reinforcement learning." arXiv preprint arXiv:1611.02779 (2016).
>
> > Try some other RNN architecture lille PFRNNs or DRBPNs
>
> As suggested by the reviewer, we ran an additional experiment using the PFRNN architectures (magenta curves in Appendix F: "Ablation Study on RNN Architectures"). Because the PFRNN is much more compute-intensive than a standard GRU, we had to decrease the number of gradient updates per environment interaction so that the experiment would finish within the rebuttal period. In most cases, the PF-GRU performs worse than a regular GRU. We suspect PF-GRU needs more hyperparameter tuning and training samples to converge.
>
> > The order of RNN and feature extraction, Layer-normalized RNN
>
> We believe these techniques are important to RNN architectures and can help recurrent policies if well-tuned. Due to the time limit, we are not able to try these techniques. We will leave them as future work.
>
> > Hypothesis about shared vs separate RNN … depends on using a dense reward
>
> We believe there may be some confusion here. The task used for this experiment (Fig. 5 of Sec. 5.2) uses a *sparse* reward (see details for the Semi-Circle environment in Appendix D.2). We believe that this experiment provides empirical evidence that the hypothesis holds for sparse reward tasks.

---

> > ### Author Response · Authors · 2021-11-24
> > **Have the revisions addressed the reviewer's concerns?**
> >
> > Dear Reviewer,
> >
> > We hope that you've had a chance to read our response. We would really appreciate a reply as to whether our revisions and new experiments have addressed the issues raised in the review, or whether there is anything else we can address.

---

> > > ### Comment · Reviewer_3HH1 · 2021-11-25
> > > **Follow-up**
> > >
> > > Sorry about the late response, I had read your response and needed to take a look at the revisions
> > >
> > > > Thank the reviewer for reminding us of the grammatical mistakes. We have fixed them in the new version.
> > > > Thank you for the suggestion. In the new version, we have replaced the bar charts with learning curves (Fig 2,3,4).
> > >
> > > Thank you, these look good!
> > >
> > > > Sorry for the confusion. Actually, our recurrent model-free RL learns well in Cheetah-Vel and is on par with Oracle, see Fig. 2. The reward is negative in Cheetah-Vel which causes confusion.
> > >
> > > Got it, the update to the graphs makes this more clear.
> > >
> > > > We have added both baselines, namely Oracle as optimal upper bound, Random as a lower bound in all the learning curves. The results of Oracle policies are from maximum of TD3 and SAC performance per task, trained with access to the hidden states (thus the POMDP becomes MDP), and with same hyperparameters as our recurrent RL.
> > >
> > > > We found there is a gap between our best variant and Oracle in most tasks, which is understandable. In some tasks such as generalization in RL and robust RL, we found given same environment and gradient step updates, Oracle performs worse than our best variant of recurrent RL. We hypothesize that this is caused by some implicit regularization performed by recurrent architectures; we leave further investigation to future work.
> > >
> > > These definitely helped make things clear for me, thank you! Interesting discovery...the regularization consideration seems intuitive. For generalization, I could see the RNN imposing a constraint on how quickly the particular parameters of the episode are inferred, perhaps making the features used by actor and critic more stable...depending on the number of tasks and how the hidden state is provided, the learning problem could also be harder for the non-recurrent version. For robust RL, I think a similar thing might apply: it's probably easier to cause a disaster by being overconfident based on hidden state than to use an initially conservative policy based on a slower update of hidden state.
> > >
> > > > We believe that the comparisons in the paper are fair: all methods are compared using the same metric (return vs environment steps) under the same assumptions. Table 6 in Appendix B clarifies the assumptions of all methods. Our paper builds on prior work on POMDPs that focuses on sample efficiency [1] and that acknowledge that agent’s past history includes observations, actions, rewards, and terminal flags [1,2].
> > >
> > > [1] Dorfman, Ron, Idan Shenfeld, and Aviv Tamar. "Offline Meta Learning of Exploration." arXiv e-prints (2020).
> > >
> > > [2] Duan, Yan, et al. "Rl^2: Fast reinforcement learning via slow reinforcement learning." arXiv preprint arXiv:1611.02779 (2016).
> > >
> > > I think the metrics are generally fine, but there's a fundamental difference between the sample efficiency of offline methods (like yours, modified VariBAD, VRM) and online methods (like MRPO, EPOpt-PPO-FF). If those are the methods you want to compare against in those environments, wouldn't basing your experiments on PPO be more appropriate?
> > >
> > > > As suggested by the reviewer, we ran an additional experiment using the PFRNN architectures (magenta curves in Appendix F: "Ablation Study on RNN Architectures"). Because the PFRNN is much more compute-intensive than a standard GRU, we had to decrease the number of gradient updates per environment interaction so that the experiment would finish within the rebuttal period. In most cases, the PF-GRU performs worse than a regular GRU. We suspect PF-GRU needs more hyperparameter tuning and training samples to converge.
> > >
> > > > We believe these techniques are important to RNN architectures and can help recurrent policies if well-tuned. Due to the time limit, we are not able to try these techniques. We will leave them as future work
> > >
> > > Fair enough, and thank you for the additional experiments!
> > >
> > > > We believe there may be some confusion here. The task used for this experiment (Fig. 5 of Sec. 5.2) uses a sparse reward (see details for the Semi-Circle environment in Appendix D.2). We believe that this experiment provides empirical evidence that the hypothesis holds for sparse reward tasks.
> > >
> > > Figure 5b is exactly what I was looking for, thank you!

---

> > > > ### Author Response · Authors · 2021-11-25
> > > > **Thank you for the positive feedback**
> > > >
> > > > First, thank you very much for the positive feedback!
> > > >
> > > > > Compare on-policy specialized methods with recurrent PPO, not recurrent TD3/SAC
> > > >
> > > > Thanks for this suggestion. The paper already includes an on-policy recurrent model-free baseline (recurrent A2C) in generalization in RL; see the yellow "A2C-RC" lines in Figure 4 and 11. We also attach the recurrent PPO (PPO-RC) in the tables below, copied from the Table 7 and 8 of [1]. In fact, recurrent PPO (PPO-RC rows) are much worse than EPOPT-PPO-FF and ours in both interpolation and extrapolation.
> > > >
> > > > Specifically, in Cheetah-Generalize, the final success rates (%) are:
> > > >
> > > > | Algorithm | Interpolation | Extrapolation |
> > > > | --- | --- | --- |
> > > > | A2C-RC | 74.70 |  42.96 |
> > > > | PPO-RC | 21.08 |  7.55 |
> > > > | EPOPT-PPO-FF| 99.28 | 53.41 |
> > > > | Ours | 98.5 | 65.3 |
> > > >
> > > > In Hopper-Generalize,
> > > >
> > > > | Algorithm | Interpolation | Extrapolation |
> > > > | --- | --- | --- |
> > > > | A2C-RC | 10.38 | 1.31 |
> > > > | PPO-RC | 0.0 |  0.0 |
> > > > | EPOPT-PPO-FF| 80.78 | 21.39 |
> > > > | Ours | 76.6 | 29.2 |
> > > >
> > > > Our recurrent model-free RL performs much better than on-policy ones (A2C-RC and PPO-RC).
> > > >
> > > > Update: We have added **recurrent PPO** (using default setting in https://github.com/ikostrikov/pytorch-a2c-ppo-acktr-gail) as an additional baseline for robust RL. We train recurrent PPO for 15M environment steps, and report its performance at 3M and 15M steps below. Note that our recurrent model-free RL was trained with 3M steps, and MRPO was trained with over 15M steps. We report both average and worst returns with std error.
> > > >
> > > > In Cheetah-Robust,
> > > >
> > > > | Algo (Env steps) | Avg Return | Worst Return |
> > > > | --- | --- | --- |
> > > > | MRPO (15M) | 223 $\pm$ 100 | 22 $\pm$ 8 |
> > > > | Recurrent PPO (15M) | 2365 $\pm$ 870 | 1414 $\pm$ 814 |
> > > > | Recurrent PPO (3M) | 2018 $\pm$ 748 | 809 $\pm$ 899 |
> > > > | **Ours (3M)** | 2277 $\pm$ 454 | 1587 $\pm$ 355 |
> > > > | Oracle (3M) | 1372 $\pm$ 518 | 896 $\pm$ 264 |
> > > >
> > > > In Hopper-Robust,
> > > >
> > > > | Algo (Env steps) | Avg Return | Worst Return |
> > > > | --- | --- | --- |
> > > > | MRPO (20M) | 1302 $\pm$ 34 | 339 $\pm$ 30 |
> > > > | Recurrent PPO (15M) | 2382 $\pm$ 529 | 779 $\pm$ 320 |
> > > > | Recurrent PPO (3M) | 1827 $\pm$ 438 | 640 $\pm$ 341 |
> > > > | **Ours (3M)** | 2392 $\pm$ 126 | 1169 $\pm$ 303 |
> > > > | Oracle (3M) | 1409 $\pm$ 238 | 427 $\pm$ 87 |
> > > >
> > > > In Walker-Robust,
> > > >
> > > > | Algo (Env steps) | Avg Return | Worst Return |
> > > > | --- | --- | --- |
> > > > | MRPO (20M) | 899 $\pm$ 223 | 127 $\pm$ 26 |
> > > > | Recurrent PPO (15M) | 2094 $\pm$ 862 | 945 $\pm$ 562 |
> > > > | Recurrent PPO (3M) | 1063 $\pm$ 544 | 446 $\pm$ 467 |
> > > > | **Ours (3M)** | 1807 $\pm$ 346 | 765 $\pm$ 503 |
> > > > | Oracle (3M) | 1146 $\pm$ 645 | 545 $\pm$ 435 |
> > > >
> > > > From the tables above, we can see that our recurrent model-free RL performs much better and more stably than recurrent PPO at 3M environment steps, and is at least on par with recurrent PPO at 15M steps, showing better sample efficiency. Similarly, recurrent PPO also greatly outperforms the specialized method MRPO.
> > > >
> > > > [1] Packer, Charles, et al. “Assessing generalization in deep reinforcement learning.” arXiv preprint arXiv:1810.12282 (2018).

---

> > > > ### Author Response · Authors · 2021-11-29
> > > > **Response to Follow-up**
> > > >
> > > > Thanks for these additional suggestions.
> > > >
> > > > Our understanding is that the reviewer's remaining concern is about the comparison with on-policy recurrent model-free RL. We have run this additional experiments and included results in the tables here: https://openreview.net/forum?id=E0zOKxQsZhN&noteId=v6KykwNsZHN
> > > >
> > > > The conclusion from these experiments is that our recurrent model-free RL outperforms the on-policy one (such as recurrent PPO) with better sample efficiency.
> > > >
> > > > Do these additional experiments address the reviewer's concerns?

---

> > > > > ### Comment · Reviewer_3HH1 · 2021-11-29
> > > > > **Further comment**
> > > > >
> > > > > My main concern was that your "ours" method in graphs comparing against EpOptPPO and MRPO should default to some on-policy algorithm (like PPO/A2C) with a bunch of the tuning methods you mentioned for a reasonable comparison. I'm not surprised that a tuned off-policy method is more sample-efficient than an on-policy method, though it is surprising to see A2C-RC performing so well. I'm not clear from the results here whether PPO-RC/A2C-RC are taking advantage of all of the other changes you made or just lifted from the original repo.
> > > > >
> > > > > Either way, I don't feel this is enough of a quibble for me to reject the submission. The paper has markedly improved in thoroughness and clarity, and I was already comfortable accepting the paper beforehand. Therefore, I'm revising my score upward

---

> > > > > > ### Author Response · Authors · 2021-11-29
> > > > > > **Re: Further comment**
> > > > > >
> > > > > > We thank the reviewer for the positive comment!
> > > > > >
> > > > > > Regarding with whether PPO/A2C-RC is well-tuned, we directly use the table results from the SunBlaze paper https://arxiv.org/abs/1810.12282 . Note that their table results came from a large-scale grid search over 4 hyperparamaters (learning rate, context length, entropy coefficient, KL coefficient; see Appendix A in their paper). So we believe the PPO/A2C-RC results were well-tuned to some extent.

---

### Official Review · Reviewer_iXRL · 2021-11-02

**Correctness:** 3
**Technical Novelty And Significance:** 3
**Empirical Novelty And Significance:** 3
**Recommendation:** 6
**Confidence:** 4

**Main Review:**

Although this paper does not propose any new algorithms or theories, I think these implementation details with RNN in RL are very practically useful experiences for solving RL problems. These introduced tricks, designments, and techniques on the architecture, input space, RL algorithms, context length, etc., are all valuable conclusions for the readers.

Moreover, the perspective viewing meta RL, robust RL, and generalization or transferability in RL as POMDP problem is also very interesting and reasonable. From this view, I can connect many of these different tasks from a principled way.

In the experiments, could the authors provide details on the network structures used for different tasks? Since efficiency of the implementations are discussed and compared with prior works. Also, instead of comparing with some SOTA methods in specific tasks, would the simplest baseline, i.e., the plain network by just removing the RNN layer, be compared as a reference in the curves? This would provide a most straightforward view that how much benefit RNN brings.

For all these tasks, will stacking RNN layers be better than a single layer?

Another question is that as demonstrated in the experiments, a carefully designed RNN implementation can always result in much better performance to solve POMDP problem, does there indeed exist theoretical explanations connecting the specific architecture of RNN and the analysis based on belief in previous POMDP theories?

Overall, although this paper focuses on RNN implementations, instead of proposing new algorithms and methods, I think the contexts discussed in the paper can contribute to the practical RL domain.


**Summary Of The Paper:**

This paper revisits recurrent neural network based model-free RL methods with carefully tuning on various aspects of the learning details. Results on a large number of tasks demonstrate that with a well-tuned RNN implementation of RL methods, it is enough to achieve sufficient performance compared with many state-of-the-art competitors on various types of tasks.


**Summary Of The Review:**

The paper provides many useful implementation details for RNN based RL methods.

---

> ### Author Response · Authors · 2021-11-20
> **Authors’ Response**
>
> We thank the reviewer for their insightful suggestions. It seems like the main suggestion in the review was to run additional experiments comparing to more complex RNN architectures and adding more baselines. We have run all these experiments suggested by the reviewer (see Sec 5.1 and Appendix E.1 and F).  **Do these new experiments address the reviewer's concerns about the experiments and baselines?**
>
> > Network Structures
>
> In the new version, we show the training details including network structures and hyperparameters in Appendix B: Training Details (Fig. 6 and Table 5).
>
> > Markovian policies as simple baselines
>
> In the new version, we show all the learning curves (Sec 5.1 and Appendix E.1) and bar charts (Fig. 1 and Fig. 18) with this simple baseline, i.e., Markovian policies trained on POMDPs, labeled as **Markovian**. To make the Markovian baseline as strong as possible, the results of Markovian policies are from maximum of TD3 and SAC performance per task. As expected, our recurrent model-free RL greatly outperforms Markovian policies in most tasks.
> Training details of our recurrent RL and Markovian policies can be also seen in Appendix B.
>
> > Stacked RNN
>
> As suggested by the reviewer, we ran an additional experiment studying whether more complex RNN architectures, such as a 2-layer stacked RNN, improve performance. In most tasks, we found that stacking RNN layers cannot outperform a single layer (see brown curves in Appendix F). We suspect that these more complex architectures may be less data-efficient to train. Please see Appendix F for the details and results of this new experiment.
>
> > Theory to explain the connection between RNN architectures and belief in POMDP literature?
>
> While there exists some literature on novel RNN architectures for POMDPs [1-6] that focuses on different generative and discriminative models, there is little theory on which inductive bias on RNN architectures can learn a better belief representation on a certain POMDP task. This is a challenging and important direction, but is beyond the scope of this empirical work.
>
> [1] Karl, Maximilian, et al. "Deep variational bayes filters: Unsupervised learning of state space models from raw data." arXiv preprint arXiv:1605.06432 (2016).
>
> [2] Karkus, Peter, David Hsu, and Wee Sun Lee. "Qmdp-net: Deep learning for planning under partial observability." arXiv preprint arXiv:1703.06692 (2017).
>
> [3] Igl, Maximilian, et al. "Deep variational reinforcement learning for POMDPs." International Conference on Machine Learning. PMLR, 2018.
>
> [4] Gregor, Karol, et al. "Shaping belief states with generative environment models for rl." arXiv preprint arXiv:1906.09237 (2019).
>
> [5] Ma, Xiao, et al. "Discriminative particle filter reinforcement learning for complex partial observations." arXiv preprint arXiv:2002.09884 (2020).
>
> [6] Wang, Yuhui, and Xiaoyang Tan. "Deep Recurrent Belief Propagation Network for POMDPs." Proceedings of the AAAI Conference on Artificial Intelligence. Vol. 35. No. 11. 2021.

---

> > ### Author Response · Authors · 2021-11-24
> > **Have the revisions addressed the reviewer's concerns?**
> >
> > Dear Reviewer,
> >
> > We hope that you've had a chance to read our response. We would really appreciate a reply as to whether our revisions and new experiments have addressed the issues raised in your review, or whether there is anything else we can address.

---

> > > ### Comment · Reviewer_iXRL · 2021-11-30
> > > **Replay to authors**
> > >
> > > Dear authors,
> > >
> > > Thanks for the reply and the revision. Most of my questions have been addressed by the new experiments and I am convinced with these new results. I believe that an appropriate RNN structure can achieve good performance in most of POMDP tasks. I do not have any further questions and I still think the work is acceptable.

---

> > > > ### Author Response · Authors · 2021-11-30
> > > > **Thank you**
> > > >
> > > > Thank you for the positive feedback!

---

### Public Comment · ~Luisa_Zintgraf1 · 2021-11-10
**Specialized methods outperform recurrent model-free RL on challenging domains**

Dear authors,

I’m the first author of the variBAD paper, the Meta-RL method compared to in this paper. I enjoyed reading your paper and think the Meta-RL community can very much benefit from a strong recurrent model-free baseline.

However, I also want to make sure that the reviewers are aware that our experiments have shown that the strength of specialized methods really only emerges in more complex tasks than were considered in this paper.

On simpler domains, including the Cheetah-Vel environment used in this paper, we found that RL$^2$ (the recurrent model-free baseline we compare to, by [Duan et al. 2016](https://arxiv.org/abs/1611.02779); [Wang et al. 2016](https://arxiv.org/abs/1611.05763)) performs similarly to variBAD. But variBAD significantly outperforms RL$^2$ on environments like Ant-Dir / Walker / Humanoid [1] or the Meta-World ML1 benchmark [2]. So while model-free methods are indeed a strong baseline, the evidence suggests that specialized algorithms are likely to outperform recurrent model-free RL on challenging benchmarks, a fact which is not reflected in this paper.

I also want to flag that the Cheetah-Vel results in this present paper are inconsistent with ours:
- Our (on-policy) implementation of variBAD achieves around -25 in the first episode. Since the results in this paper show performance over 2 episodes (according to the configs in the provided source code), this would equate to -50. These results are based on PPO and obtained after 100M frames.
- The (off-policy) results in this paper show around -85 for RL$^2$ and -175 for VariBAD. These results are based on SAC and obtained after 5M frames.

This paper suggests that off-policy variBAD is “stronger” than on-policy variBAD. However, the obtained results are worse than published results in terms of final performance. Our results on Cheetah-Vel can be obtained using the code at [https://github.com/lmzintgraf/varibad](https://github.com/lmzintgraf/varibad). It would be great if the authors could address these discrepancies.

[1] MuJoCo results - see Fig 4 in the [variBAD paper](https://arxiv.org/pdf/1910.08348.pdf), or improved and additional results [here](https://www.dropbox.com/s/46if96420a6o96w/mujoco_test_performance.png). These can be reproduced with the code at [https://github.com/lmzintgraf/varibad](https://github.com/lmzintgraf/varibad).

[2] MetaWorld results - see Table 2 in the appendix of [this paper](https://arxiv.org/pdf/2010.01062.pdf), or the table [here](https://www.dropbox.com/s/0dvq2rd26hxjpc7/metaworld_results.png).

---

> ### Author Response · Authors · 2021-11-22
> **Authors' Response**
>
> Hi Luisa,
>
> Thank you for your interest in our work!
>
> > This paper suggests that off-policy VariBAD [2] is “stronger” than on-policy VariBAD [1]
>
> As suggested, we have reworded the sentence in Sec. 5.1 that "We make VariBAD stronger …" to clarify that the change from PPO to SAC in VariBAD may not improve the asymptotic performance.
>
> > Why on-policy VariBAD in the original paper is better than off-policy VariBAD in our paper (e.g. in Cheetah-vel) ?
>
> We believe that our comparisons with prior work are fair. To rule out the possibility of re-implementation bugs in the off-policy VariBAD, we ran the official off-policy VariBAD implementation (https://github.com/Rondorf/BOReL) on the meta RL tasks from our paper. Recurrent model-free RL also matches or outperforms this implementation of off-policy VariBAD, supporting the conclusions in the original paper.
>
> > Try harder environment like Ant-Dir / Walker / Humanoid. On-policy VariBAD outperforms recurrent model-free RL on challenging domains
>
> We thank the authors for the suggestion. Using Ant-Dir, we have compared recurrent model-free RL to off-policy VariBAD. These new results, shown in Figure 24 in Appendix G, indicate that recurrent model-free RL significantly outperforms off-policy VariBAD. We would be happy to include the results from on-policy VariBAD (Fig. 6 in [1]), if the authors would be willing to share these curves.
>
> [1] Zintgraf, Luisa, et al. "Varibad: A very good method for bayes-adaptive deep rl via meta-learning." arXiv preprint arXiv:1910.08348 (2019).
>
> [2] Dorfman, Ron, Idan Shenfeld, and Aviv Tamar. "Offline Meta Learning of Exploration." arXiv e-prints (2020): arXiv-2008.

---

### Author Response · Authors · 2021-11-20
**New version of our paper**

We thank all the reviewers for acknowledging the importance of our work as a strong recurrent model-free RL baseline for POMDPs. Based on the suggestions of the reviewers, we have run additional experiments and significantly revised the experiment section of our paper. We have added three new baselines:
1. Oracle policies that have access to the hidden states as an upper bound,
2. Markovian policies as a non-trivial lower bound, and
3. Random policies as a trivial lower bound.

Please see the experiment section (Sec 5.1) and corresponding appendix sections (B, E.1, F) for details. Paragraphs that are newly added are shown in purple. In individual responses to reviewers, we describe how we revised figures and added clarifications to the paper.

---

### Decision · Program_Chairs · 2022-01-20

**Decision:**

Reject

**Comment:**

This paper recognizes that several common sub-problems studied in RL, such as meta RL and generalization in RL, can be cast as POMDPs. Using this observation, the authors evaluate how a straightforward approach to deal with POMDPs---using a recurrent neural network---compares to more specialized approaches. The reviewers agree that the research question studied in this paper is very interesting. However, after careful deliberation, I share the view of reviewer 2WFY that the results insufficiently support the claims made in the paper. In particular, I view the main claim from the abstract "We find that a careful architecture and hyperparameter decisions yield a recurrent model-free implementation that performs on par with (and occasionally substantially better than) more sophisticated recent techniques in their respective domains."  as insufficiently supported. The main issue with the experiments is that only a small number of simple domains are considered. As Luisa points out in the public comments, variBAD dominates recurrent baselines when more complex tasks are considered, while on simpler domains such as the Cheetah-Vel domain considered in this paper, it performs similar to a recurrent model-free baseline. In the rebuttal the authors have added a more complex domain to address this, showing that a recurrent model-free baseline outperforms an off-policy version of variBAD. However, I view these results as inconclusive, as only a single complex domain is considered and they appear to contradict previous results with on-policy variBAD. For these reasons, I don't think the work in its current form is ready for publication at ICLR. But I want to encourage the authors to work out this direction further. In particular, adding more complex domains and also considering the on-policy variBAD method, can make this work stronger.